# Widespread premature transcription termination of *Arabidopsis thaliana* NLR genes by the spen protein FPA

Matthew T Parker[1†], Katarzyna Knop[1†], Vasiliki Zacharaki[1†], Anna V Sherwood[1], Daniel Tomé[2], Xuhong Yu[3], Pascal GP Martin[3‡], Jim Beynon[2], Scott D Michaels[3], Geoffrey J Barton[1], Gordon G Simpson[1,4]*

[1]School of Life Sciences, University of Dundee, Dundee, United Kingdom; [2]School of Life Sciences, University of Warwick, Coventry, United Kingdom; [3]Department of Biology, Indiana University, Bloomington, United States; [4]The James Hutton Institute, Invergowrie, United Kingdom

**Abstract** Genes involved in disease resistance are some of the fastest evolving and most diverse components of genomes. Large numbers of nucleotide-binding, leucine-rich repeat (NLR) genes are found in plant genomes and are required for disease resistance. However, NLRs can trigger autoimmunity, disrupt beneficial microbiota or reduce fitness. It is therefore crucial to understand how NLRs are controlled. Here, we show that the RNA-binding protein FPA mediates widespread premature cleavage and polyadenylation of NLR transcripts, thereby controlling their functional expression and impacting immunity. Using long-read Nanopore direct RNA sequencing, we resolved the complexity of NLR transcript processing and gene annotation. Our results uncover a co-transcriptional layer of NLR control with implications for understanding the regulatory and evolutionary dynamics of NLRs in the immune responses of plants.

***For correspondence:**
g.g.simpson@dundee.ac.uk

[†]These authors contributed equally to this work

**Present address:** [‡]INRAE, Univ. Bordeaux, Villenave d'Ornon, France

**Competing interests:** The authors declare that no competing interests exist.

## Introduction

In plants and animals, NLR (nucleotide-binding, leucine-rich repeat) proteins function to detect the presence and activity of pathogens (*Barragan and Weigel, 2020*; *Jones et al., 2016*; *Tamborski and Krasileva, 2020*). Plant genomes can encode large numbers of NLR genes, which often occur in physical clusters (*Jiao and Schneeberger, 2020*; *Wei et al., 2016*). Powerful selective pressure drives the rapid birth and death of NLR genes, resulting in intraspecific diversity in NLR alleles and gene number. Consequently, the near-complete repertoire of Arabidopsis NLR genes was only recently revealed using long-read DNA sequencing of diverse Arabidopsis accessions (*Van de Weyer et al., 2019*).

In plants, NLR proteins generally comprise an N-terminal Toll/interleukin receptor (TIR), coiled-coil (CC) or RPW8 domain that facilitates signalling; a central nucleotide-binding NB-ARC domain that acts as a molecular switch; and C-terminal leucine-rich repeats (LRRs) that interact with target proteins. NLRs can recognise pathogen effectors either directly by binding to them through LRR domains or indirectly by detecting modifications to host proteins caused by effector action. In some cases, domains of host proteins targeted by pathogen effectors have been incorporated into NLRs as integrated domains (or decoys) (*Le Roux et al., 2015*). NLRs that interact directly with effectors are under high levels of diversifying selection to modify their recognition specificities, resulting in significant allelic polymorphism (*Prigozhin and Krasileva, 2021*). Genomic variation also yields diversity in NLR protein organisation, through domain swapping or truncating mutations, and NLR isoforms that lack NB-ARC or LRR domains can function in plant immune responses (*Nishimura et al., 2015*; *Swiderski et al., 2009*; *Zhang and Gassmann, 2007*). The consequence of

this diversity is that there is no one-size-fits-all explanation of how NLR proteins function (*Barragan and Weigel, 2020*).

The benefit of NLRs to the host is disease resistance, but the costs of increased NLR diversity or activity can include detrimental autoimmunity (*Rodriguez et al., 2016*), reduced association with beneficial microbes (*Yang et al., 2010*) and a general reduction in fitness (*Tian et al., 2003*). In some cases, autoimmunity caused by epistatic interactions involving NLRs can cause hybrid necrosis (*Chae et al., 2014*). Therefore, a key question is how NLRs are regulated to enable limited expression for pathogen surveillance but enhanced expression during defence responses. This problem is compounded by the evolutionary dynamics of NLRs because regulatory processes must keep pace with the emergence of new NLR genes and gain or loss of function in others. Consequently, the regulation of NLRs is one of the most important and difficult challenges faced by plants.

NLR control measures occur at different stages of gene expression (*Lai and Eulgem, 2018*). For example, microRNAs limit the expression of many NLRs by targeting conserved regions encoded in NLR mRNAs and triggering cascades of phased siRNAs that broadly suppress NLR activity (*Cai et al., 2018*; *Canto-Pastor et al., 2019*; *Shivaprasad et al., 2012*; *Zhai et al., 2011*). Alternative splicing, which promotes the simultaneous expression of more than one NLR isoform, is required for the functions of both the *N* gene which provides resistance to tobacco mosaic virus (*Dinesh-Kumar and Baker, 2000*), and *RECOGNITION OF PSEUDOMONAS SYRINGAE 4* (*RPS4*), which confers resistance to *Pseudomonas syringae* DC3000 in Arabidopsis (*Zhang and Gassmann, 2007*). Alternative polyadenylation at intragenic heterochromatin controls the expression of Arabidopsis *RECOGNITION OF PERONOSPORA PARASITICA 7* (*RPP7*), with functional consequences for immunity against the oomycete pathogen *Hyaloperonospora arabidopsidis* (*Tsuchiya and Eulgem, 2013*). Finally, RNA surveillance pathways control NLRs. For example, null mutants defective in nonsense-mediated RNA decay (NMD) are lethal in Arabidopsis because they trigger NLR *RPS6*-dependent autoimmunity (*Gloggnitzer et al., 2014*). Conversely, mutations in the RNA exosome, which degrades RNAs in a 3′ to 5′ direction, suppress *RPS6*-dependent autoimmune phenotypes (*Takagi et al., 2020*). Fine tuning of different levels of NLR control may be integrated to produce quantitative patterns of disease resistance (*Corwin and Kliebenstein, 2017*), but our understanding of how this occurs globally is fragmentary and incomplete (*Adachi et al., 2019*).

The RNA-binding protein FPA was first identified as a factor required for the control of Arabidopsis flowering time (*Koornneef et al., 1991*). Loss-of-function *fpa* mutants flower late due to elevated levels of the floral repressor, FLC (*Schomburg et al., 2001*). However, this cannot be the only function of FPA because it is much more widely conserved than *FLC*. FPA is a member of the spen family of proteins, which are defined by three N-terminal RNA recognition motifs and a C-terminal protein interaction SPOC domain (*Ariyoshi and Schwabe, 2003*). We previously showed that FPA controls the site of cleavage and polyadenylation in some mRNAs, including autoregulation of *FPA* pre-mRNA (*Duc et al., 2013*; *Hornyik et al., 2010*; *Lyons et al., 2013*). These findings were extended to show that FPA can affect poly(A) site choice at genes with intronic heterochromatin, including *RPP7* (*Deremetz et al., 2019*). The poly(A) site selection mechanism used by FPA remains unclear. FPA might mediate poly(A) site choice either directly by recruiting the RNA 3′ end processing machinery to sensitive sites or indirectly, for example by influencing splicing, chromatin modifications or the rate of transcription by RNA Polymerase II (Pol II). We previously used Helicos Biosciences direct RNA sequencing (Helicos DRS) to map the 3′ ends of Arabidopsis polyadenylated transcripts and identify genes affected by transcriptome-wide loss of FPA function (*Duc et al., 2013*; *Sherstnev et al., 2012*). A limitation of this approach was that it could only identify RNA 3′ end positions, and so could not resolve other potential roles of FPA in gene expression.

In this study, we used two approaches to gain a clearer understanding of how FPA functions. We first investigated which proteins FPA associates with inside living plant cells. Next, we analysed the global impact of different levels of FPA activity on gene expression. For this, we combined Helicos DRS with short-read Illumina RNA-Seq and Oxford Nanopore Technologies (Nanopore) DRS, which can reveal the authentic processing and modification of full-length mRNAs (*Parker et al., 2020*). Using these combined data together with new computational approaches to study RNA processing, we found that the predominant role of FPA is to promote poly(A) site choice. In addition, we uncovered an unusual degree of complexity in the processing of NLR mRNAs, which is sensitive to FPA. The finding that premature transcription termination functions as an additional layer of NLR expression control has implications for understanding the dynamics of NLR regulation and evolution.

## Results

### FPA co-purifies with proteins that mediate mRNA 3′ end processing

In order to understand how FPA controls the site of mRNA 3′ end formation, we used in vivo interaction proteomics–mass spectrometry (*IVI-MS*) to identify which proteins FPA associates with inside living plant cells. First, we fixed molecular interactions using formaldehyde infiltration of Arabidopsis seedlings expressing FPA fused to YFP (*35S::FPA:YFP*). Wild-type Columbia-0 (Col-0) seedlings treated in the same way were used as a negative control. We then purified nuclei and performed GFP-trap immunopurification followed by liquid chromatography–tandem mass spectrometry (LC-MS/MS) to identify FPA-associated proteins. By comparing the proteins detected in three biological replicates of *35S::FPA:YFP* and Col-0, we identified 203 FPA co-purifying proteins with a median $\log_2$ fold change in adundance of greater than two (*Figure 1—figure supplement 1*). At least 56% (113) of the enriched proteins are poly(A)+mRNA binding proteins as established by orthogonal RNA-binding proteome analysis (*Bach-Pages et al., 2020*; *Reichel et al., 2016*).

Consistent with FPA control of mRNA 3′ end formation, 14 highly conserved cleavage and polyadenylation factors (CPFs) co-purified with FPA (*Figure 1A*, *Supplementary file 1*). These include members of the cleavage and polyadenylation specificity factor (CPSF) complex, cleavage stimulating factor (CstF) complex, and cleavage factor I and II (CFIm/CFIIm) complexes. The U2AF and U2 spliceosome components that interact with CFIm–CPSF to mediate terminal exon definition were also detected (*Kyburz et al., 2006*; *Figure 1B*, *Supplementary file 1*). We additionally detected both subunits of Pol II. Characteristically, Serine[5] of the Pol II C-terminal domain (CTD) heptad repeat is phosphorylated when Pol II is at the 5′ end of genes, and Ser[2] is phosphorylated when Pol II is at the 3′ end (*Komarnitsky et al., 2000*). The position-specific phosphorylation of these sites alters the RNA processing factors which are recruited to the CTD at the different stages of transcription. We found that the kinase CDKC;2, which phosphorylates Ser[2] (*Wang et al., 2014*), and the phosphatase CPL1 (homolog of yeast Fcp1), which dephosphorylates Ser[5] (*Koiwa et al., 2004*), co-purified with FPA. We also detected the homolog of the human exonuclease XRN2 (known as XRN3 in Arabidopsis), which mediates Pol II transcription termination (*Krzyszton et al., 2018*).

A second major class of proteins that co-purified with FPA are components of the autonomous flowering pathway (*Andrés and Coupland, 2012*; *Simpson, 2004*; *Figure 1C*, *Supplementary file 1*). FPA functions in the autonomous pathway to limit expression of the floral repressor *FLC*. FPA activity is associated with alternative polyadenylation of long non-coding RNAs that are transcribed antisense to the *FLC* locus (*Hornyik et al., 2010*; *Liu et al., 2007*). Consistent with this, conserved CPF proteins such as FY (WDR33) (*Simpson et al., 2003*), PCFS4 (*Xing et al., 2008*), CSTF64 and CSTF77 (*Liu et al., 2010*) were previously identified in late flowering mutant screens. Other detected autonomous pathway factors are proteins with established roles in pre-mRNA processing, including HLP1 (*Zhang et al., 2015*), FLK (*Mockler et al., 2004*) and EMB1579/RSA1 (*Zhang et al., 2020b*). Notably, FLK has been found to associate with PEP, HUA1, and HEN4 (*Zhang et al., 2015*), and we identified all four of these as FPA co-purifying proteins. In addition to regulating *FLC*, the FLK–PEP complex has been shown to control alternative polyadenylation within pre-mRNA encoding the floral homeotic transcription factor AGAMOUS (*Rodríguez-Cazorla et al., 2015*). Their co-purification with FPA suggests that this role may be more global and involve direct interactions at RNA 3′ ends.

A third group of proteins that co-purified with FPA are conserved members of the mRNA N[6]-methyladenosine (m[6]A) writer complex (*Růžička et al., 2017*; *Figure 1D*, *Supplementary file 1*). The m[6]A modification mediated by this complex is predominately targeted to the 3′ untranslated region (UTR) of Arabidopsis protein-coding mRNAs (*Parker et al., 2020*). The co-purification of FPA with m[6]A writer complex components may be explained by either a direct role for FPA in m[6]A modification or, more simply, because both CPF and m[6]A writer proteins are found at RNA 3′ ends.

The picture that emerges from this analysis is that FPA is located in proximity to proteins that promote cleavage, polyadenylation, transcription termination and RNA modification at the 3′ end of Pol II-transcribed genes.

### FPA co-localises with RNA Pol II Ser[2] at the 3′ end of Arabidopsis genes

We next used an orthogonal approach to investigate the association of FPA with proteins that function at the 3′ end of Pol II-transcribed genes. We performed chromatin immumunoprecipitation

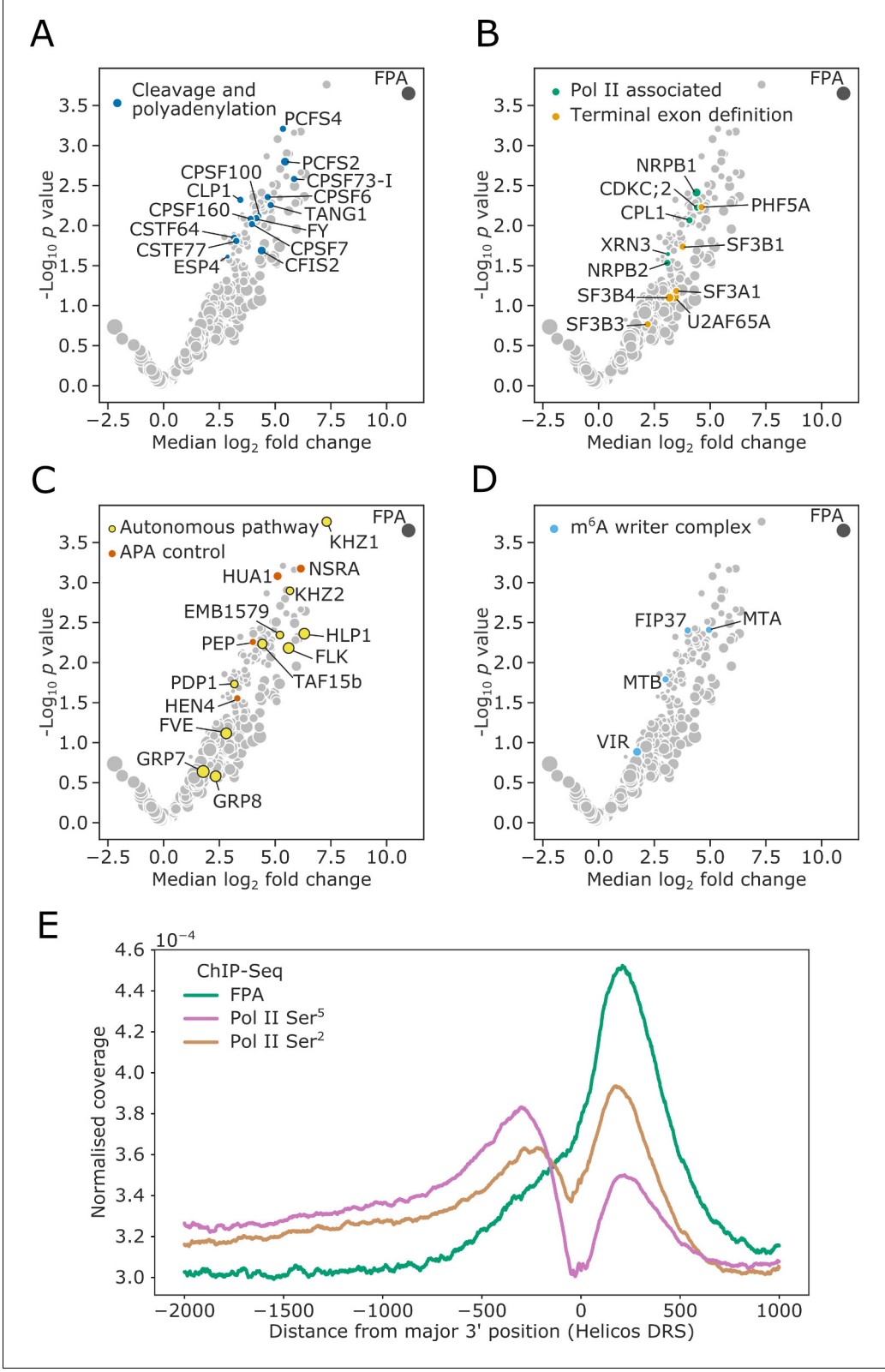

**Figure 1.** FPA associates with proteins that function to process the 3′ ends of Pol II-transcribed RNAs and promote transcription termination. (**A–D**) Volcano plots representing proteins co-purifying with FPA using *IVI-MS*. Only proteins detected in all three biological replicates of the *35S::FPA:YFP* line are shown (light grey). The following classes are highlighted: (**A**) CPFs in dark blue; (**B**) Pol II-associated factors in green; terminal exon

*Figure 1 continued on next page*

*Figure 1 continued*

definition factors in dark orange; (C) autonomous pathway components in yellow and factors controlling alternative polyadenylation in light orange; and (D) m$^6$A writer complex components in light blue. (E) ChIP-Seq metagene profile showing the normalised occupancy of FPA (green) and Pol II phosphorylated at Ser$^5$ (pink) and Ser$^2$ (brown) of the CTD (*Yu et al., 2019*) relative to the major 3′ position of each gene, as measured using Helicos DRS. Only long genes (>2.5 kb) are included (*n* = 10,215).

The online version of this article includes the following figure supplement(s) for figure 1:

**Figure supplement 1.** FPA co-localises with Pol II Ser$^2$ at the 3′ end of genes.

**Figure supplement 2.** FPA controls Pol II occupancy and chimeric RNA formation at *PIF5*.

sequencing (ChIP-Seq) using antibodies against FPA and Pol II phosphorylated at either Ser$^5$ or Ser$^2$ of the CTD heptad repeat (*Yu et al., 2019*). Our metagene analysis revealed that FPA is enriched at the 3′ end of genes and co-localises with Pol II phosphorylated at Ser$^2$ of the CTD (*Figure 1E*, *Figure 1—figure supplement 1*). We found that FPA occupancy at 3′ ends was well correlated with Pol II Ser$^2$ occupancy (Spearman's ρ = 0.67, p<2 × 10$^{-308}$, 95% confidence interval [0.66, 0.68]). The close relationship between FPA and Pol II Ser$^2$ is reinforced by changes in the distribution of Pol II isoforms in *fpa* mutants. For example, we previously showed that FPA is required for 3′ end processing at *PIF5* (*Duc et al., 2013*). Pol II Ser$^2$ was enriched at the 3′ end of *PIF5* in Col-0 but depleted from this region in *fpa-7* mutants (*Figure 1—figure supplement 2*). Together, these orthogonal ChIP-Seq and *IVI-MS* analyses reveal the close association of FPA with proteins involved in 3′ end processing and transcription termination at the 3′ end of Arabidopsis genes.

## FPA predominantly promotes poly(A) site choice

We next asked which RNA processing events are controlled by FPA. We used a combination of Illumina RNA-Seq and Helicos and Nanopore DRS technologies to analyse three different genetic backgrounds expressing different levels of FPA activity: wild-type Col-0, loss-of-function *fpa-8* and a line overexpressing FPA fused to YFP (*35S::FPA:YFP*). In combination, these orthogonal sequencing technologies can reveal different features of transcriptomes: Helicos DRS short reads identify the 3′ ends of mRNAs, but cannot reveal the full properties of the corresponding transcripts (*Ozsolak et al., 2009*) Illumina RNA-Seq produces short reads derived from all expressed regions, meaning that changes in RNA 3′ end processing can only be detected by differences in coverage (*Xia et al., 2014*) and Nanopore DRS long reads define the 3′ ends of mRNAs in the context of reads that can correspond to full-length transcripts (*Parker et al., 2020*). For each genotype, we performed three biological replicates with Helicos DRS, six with Illumina RNA-Seq and four with Nanopore DRS. The resultant sequencing statistics are detailed in *Supplementary file 1*.

We first assessed the utility of the three sequencing technologies to map changes in mRNA processing by focusing on the *FPA* locus. FPA autoregulates its expression by promoting premature cleavage and polyadenylation within intron 1 of *FPA* pre-mRNA (*Duc et al., 2013*; *Hornyik et al., 2010*). Consistent with this, a proximal poly(A) site in the first intron and distal sites in the terminal intron and exon of *FPA* could be mapped in Col-0 using Nanopore and Helicos DRS (*Figure 2A*). Using all three data types, we detected a quantitative shift towards selection of distal poly(A) sites in the loss-of-function *fpa-8* mutant and a strong shift to proximal poly(A) site selection when FPA is overexpressed (*35S::FPA:YFP*). Nanopore DRS provided the clearest picture of alternative polyadenylation events because full-length reads reveal poly(A) site choice in the context of other RNA processing events.

We next asked how transcriptome-wide RNA processing is affected by FPA activity. Since mutations in FPA cause readthrough of annotated 3′UTRs (*Duc et al., 2013*), we applied the software tool StringTie2 (*Pertea et al., 2015*) to create a bespoke reference annotation with Nanopore DRS reads from Col-0, *fpa-8* and *35S::FPA:YFP*. We then measured how changes in FPA expression altered the 3′ end distribution at each locus using the earth mover's distance (EMD; also known as the Wasserstein distance). EMD indicates the 'work' required to transform one normalised distribution into another based on the proportion of 3′ ends that would have to be moved and by what distance. We used an EMD permutation test, in which reads are randomly shuffled between conditions,

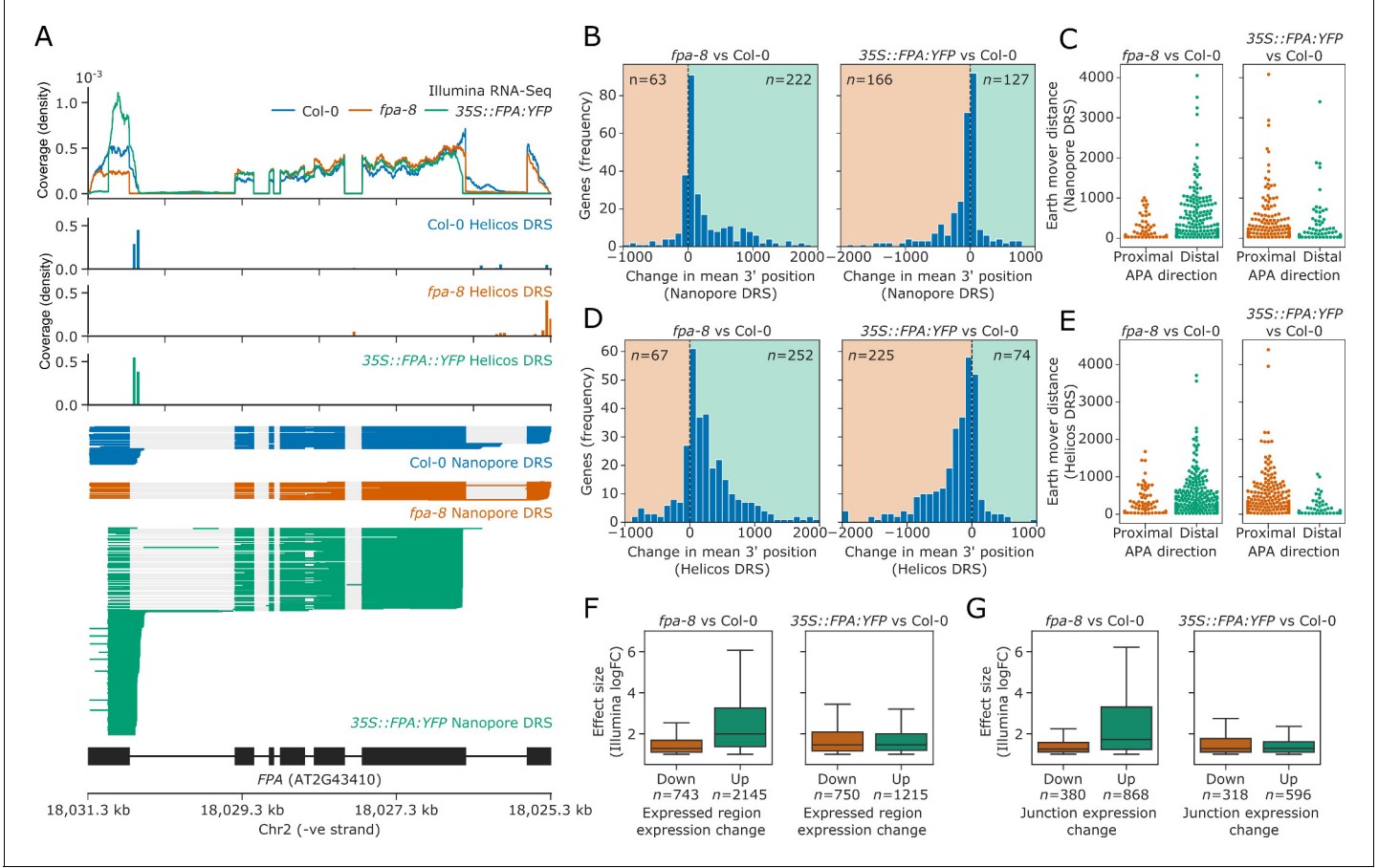

**Figure 2.** FPA-dependent poly(A) site selection. Loss of FPA function is associated with the preferential selection of distal poly(A) sites, whereas FPA overexpression leads to the preferential selection of proximal poly(A) sites. (**A**) Illumina RNA-Seq, Helicos DRS and Nanopore DRS reveal FPA-dependent RNA 3′ end processing changes at the *FPA* (*AT2G43410*) locus. The *35S::FPA:YFP* construct has alternative transgene-derived untranslated regions, so mRNAs derived from the transgene do not align to the native FPA 5′UTR and 3′UTR. (**B**) Histograms showing change in mean RNA 3′ end position for significantly alternatively polyadenylated loci (EMD >25, FDR < 0.05) in *fpa-8* (left panel) and *35S::FPA:YFP* (right panel) compared with Col-0, as detected using Nanopore DRS. Orange and green shaded regions indicate sites with negative and positive RNA 3′ end position changes, respectively. (**C**) Effect size of significant proximal (orange) and distal (green) alternative polyadenylation events in *fpa-8* (left panel) and *35S::FPA:YFP* (right panel) compared with Col-0, as measured using the EMD. (**D**) Histograms showing change in mean RNA 3′ end position for significantly alternatively polyadenylated loci (EMD >25, FDR < 0.05) in *fpa-8* (left panel) and *35S::FPA:YFP* (right panel) compared with Col-0, as detected using Nanopore DRS. Orange and green shaded regions indicate sites with negative and positive RNA 3′ end position changes, respectively. (**E**) Effect size of significant proximal (orange) and distal (green) alternative polyadenylation events in *fpa-8* (left panel) and *35S::FPA:YFP* (right panel) compared with Col-0, as measured using the EMD. (**F**) Boxplots showing the effect size (absolute log₂ fold change (logFC)) of alternatively processed loci identified using Illumina RNA-Seq in *fpa-8* (left panel) and *35S::FPA:YFP* (right panel) respectively. Down- and upregulated loci are shown in orange and green, respectively. For each locus, the region with the largest logFC was selected to represent the locus. Loci with both up- and downregulated regions contribute to both boxes. (**G**) Boxplots showing the effect size (absolute logFC) of loci with alternative splice junction usage identified using Illumina RNA-Seq in *fpa-8* (left panel) and *35S::FPA:YFP* (right panel), respectively. Down- and upregulated loci are shown in orange and green, respectively. For each locus, the junction with the largest logFC was selected to represent the locus. Loci with both up- and downregulated junctions contribute to both boxes.

The online version of this article includes the following source data and figure supplement(s) for figure 2:

**Source data 1.** Nanopore StringTie assembly [Linked to *Figure 2A–B*].
**Source data 2.** Differential 3′ processing results for *fpa-8* vs Col-0, as identified by Nanopore DRS [Linked to *Figure 2B–C*].
**Source data 3.** Differential 3′ processing results for *35S::FPA:YFP* vs Col-0, as identified by Nanopore DRS [Linked to *Figure 2B–C*].
**Source data 4.** Differential 3′ processing results for *fpa-8* vs Col-0, as identified by Helicos DRS [Linked to *Figure 2D–E*].
**Source data 5.** Differential 3′ processing results for *35S::FPA:YFP* vs Col-0, as identified by Helicos DRS [Linked to *Figure 2D–E*].
**Source data 6.** Differentially expressed regions results for *fpa-8* vs Col-0, as identified by Illumina RNA-Seq [Linked to *Figure 2F*].
**Source data 7.** Differentially expressed regions results for *35S::FPA:YFP* vs Col-0, as identified by Illumina RNA-Seq [Linked to *Figure 2F*].
**Source data 8.** Differential splice junction usage results for *fpa-8* vs Col-0, as identified by Illumina RNA-Seq [Linked to *Figure 2G*].

*Figure 2 continued on next page*

*Figure 2 continued*

**Source data 9.** Differential splice junction usage results for *35S::FPA:YFP* vs Col-0, as identified by Illumina RNA-Seq [Linked to *Figure 2G*].
**Figure supplement 1.** Nanopore and Helicos DRS reveal FPA-dependent RNA 3′ end processing changes.
**Figure supplement 2.** Splicing alterations in *fpa-8* can be explained by changes in RNA 3′ end formation.
**Figure supplement 3.** FPA does not affect global mRNA m$^6$A methylation.
**Figure supplement 3—source data 1.** m$^6$A : A ratios for Col-0, *fpa-8*, *35S::FPA:YFP* and *vir-1*, as detected by LC-MS/MS [Linked to *Figure 2—figure supplement 3*].
**Figure supplement 4.** FPA-dependent control of NLR expression is independent of IBM1.
**Figure supplement 4—source data 1.** Differential H3K9me$^2$ results for *ibm1–four* vs Col-0 [Linked to *Figure 2—figure supplement 4*].

to estimate *p*-values for each locus. Loci with an EMD greater than 25 and a false discovery rate (FDR) less than 0.05 were considered differentially polyadenylated.

Using this approach on Nanopore DRS data, we identified 285 and 293 loci with alternative polyadenylation events in *fpa-8* and *35S::FPA:YFP*, respectively (*Figure 2B*). In all, 77.9% (222) of loci with alternative polyadenylation in *fpa-8* displayed a positive change in the mean 3′ end position, indicating a predominant shift to distal poly(A) site selection (*Figure 2B*, left panel). These loci also had greater effect sizes than those with shifts towards proximal poly(A) sites (*Figure 2C*, left panel). In contrast, 56.7% (166) of loci with alternative polyadenylation in *35S::FPA:YFP* displayed a negative change in the mean 3′ end position, indicating a shift towards proximal poly(A) sites (*Figure 2B*, right panel). These loci had greater effect sizes than those with positive changes in 3′ end profile (*Figure 2C*, right panel). A total of 16 loci displayed a shift to distal poly(A) site selection in *fpa-8* and to proximal poly(A) site selection in *35S::FPA:YFP* (hypergeometric test p=3.9 × 10$^{-7}$), demonstrating that loss of function versus overexpression of FPA can result in reciprocal patterns of poly(A) site choice.

We used the same approach to identify loci with FPA-dependent alternative polyadenylation in Helicos DRS data. We identified 319 and 299 genes with alternative polyadenylation events in *fpa-8* and *35S::FPA:YFP*, respectively (*Figure 2D and E*). Consistent with Nanopore DRS analysis, the predominant shifts in *fpa-8* and *35S::FPA:YFP* were towards distal (79.0% or 252 loci) and proximal (75.3% or 225 loci) poly(A) sites, respectively. In all, 44 loci displayed a shift to distal poly(A) sites in *fpa-8* and to proximal poly(A) sites in *35S::FPA:YFP* (hypergeometric test p=4.8 × 10$^{-30}$), again demonstrating reciprocal poly(A) site selection depending on FPA activity. Of the 222 loci identified with shifts to distal poly(A) sites in *fpa-8* using Nanopore DRS, 39.6% (88) were also detected using Helicos DRS (*Figure 2—figure supplement 1*). Likewise, 44.0% of loci (73) with proximal polyadenylation detected in *35S::FPA:YFP* using Nanopore DRS were also detected using Helicos DRS. Across the DRS datasets, we identified 59 loci for which reciprocal poly(A) site selection depending on FPA activity could be detected by Nanopore DRS and/or Helicos DRS.

In order to analyse the Illumina RNA-Seq data, we developed annotation-agnostic software for detecting alternative RNA 3′ end processing events, using a similar approach to the existing tools DERfinder (*Collado-Torres et al., 2017*), RNAprof (*Tran et al., 2016*), and DEXSeq (*Anders et al., 2012*). We segmented Illumina RNA-Seq data by coverage and relative expression in *fpa*-8 or *35S::FPA:YFP* compared with Col-0. Segmented regions were grouped into transcriptional loci using the annotations generated from Nanopore DRS datasets. Differential usage of regions within each locus was then tested using DEXSeq. Using this approach, we identified 2535 loci with differential RNA processing events in *fpa-8*: 1792 were upregulated, 390 were downregulated, and 353 had both upregulated and downregulated regions (FDR < 0.05, absolute logFC >1; *Figure 2F*, left panel). A total of 1747 loci with differential RNA processing events were identified in *35S::FPA:YFP*: 997 were upregulated, 532 were downregulated, and 218 had both upregulated and downregulated regions (*Figure 2F*, right panel). The median effect size for differentially processed regions was greater for upregulated regions than for downregulated regions in *fpa-8*. This is consistent with an increase in transcriptional readthrough events and elevated expression of intergenic regions and downstream genes. In contrast, the median effect size for differentially processed regions was similar for up- and downregulated regions in *35S::FPA:YFP*. This is consistent with an increase in the relative expression of proximal exonic and intronic regions, and loss of expression of distal exonic regions caused by

preferential selection of proximal poly(A) sites. Similar results were seen for differential splice junction usage analysis (*Figure 2G*), suggesting that changes in splicing are the indirect effects of altered 3′ end processing in *fpa-8*, rather than direct effects of FPA on splice site choice. Evidence of this can be seen at the *PIF5* locus, where readthrough results in increased cryptic and canonical splicing of downstream *PAO3* (*Figure 2—figure supplement 2*).

We next asked whether FPA influences RNA modification. Our *IVI-MS* analysis had revealed that conserved members of the Arabidopsis m$^6$A writer complex co-purify with FPA (*Figure 1D*, *Supplementary file 1*). The human proteins most closely related to FPA are RBM15/B, which co-purify with the human m$^6$A writer complex and are required for m$^6$A deposition (*Patil et al., 2016*). We used LC-MS/MS to analyse the m$^6$A/A (adenosine) ratio in mRNA purified from Col-0, *fpa-8*, *35S::FPA:YFP* and a mutant defective in the m$^6$A writer complex component VIR (*vir-1*). Consistent with previous reports, the level of mRNA m$^6$A in the hypomorphic *vir-1* allele was reduced to approximately 10% of wild-type levels (*Parker et al., 2020*; *Růžička et al., 2017*; *Figure 2—figure supplement 3*). However, we detected no differences in the m$^6$A level between genotypes with altered FPA activity. Therefore, we conclude that FPA does not influence global levels of mRNA m$^6$A methylation.

Finally, we asked whether the FPA-dependent global changes in alternative polyadenylation result from an indirect effect on chromatin state. We previously showed that FPA controls the expression of histone demethylase IBM1 by promoting proximal polyadenylation within *IBM1* intron 7 (*Duc et al., 2013*). IBM1 functions to restrict H3K9me$^2$ levels, and *ibm1* mutants accumulate ectopic heterochromatic marks in gene bodies, which affects RNA processing at certain loci (*Miura et al., 2009*; *Saze et al., 2008*). When we analysed two independent ChIP-Seq datasets of H3K9me$^2$ in *ibm1–4* mutants (*Inagaki et al., 2017*; *Lai et al., 2020*), we found that only 10.6% of loci with altered poly(A) site choice in *35S::FPA:YFP* have altered H3K9me$^2$ in *ibm1* mutants compared with 14.2% of all loci tested (hypergeometric p=0.97; *Figure 2—figure supplement 4*). This result suggests that FPA-dependent poly(A) site choice is not an indirect consequence of FPA control of *IBM1*.

Overall, these analyses reveal that the primary function of FPA is to control poly(A) site choice. FPA predominantly promotes poly(A) site selection; hence, *fpa* loss-of-function backgrounds exhibit readthrough at sites used in the wild type, whereas FPA overexpression results in increased selection of proximal poly(A) sites.

## NLRs are major targets of FPA-sensitive alternative poly(A) site selection

We next asked which groups of genes are sensitive to FPA-dependent alternative polyadenylation. We used InterPro annotations (*Mitchell et al., 2019*) to perform protein family domain enrichment analysis of the loci affected by FPA (revealed by the Nanopore and Helicos DRS analyses). We found that sequences encoding NB-ARC, Rx-like coiled coil (CC), and/or LRR domains were enriched amongst the loci with increased proximal polyadenylation in *35S::FPA:YFP* (*Figure 3A and B*). This combination of domains is associated with NLR disease resistance proteins.

The Col-0 accession contains at least 206 genes encoding some combination of TIR, CC, RPW8, NB-ARC, and LRR domains, which might be classified as NLRs or partial NLRs (*Van de Weyer et al., 2019*). In general, these can be grouped according to their encoded N-terminal domain as TIR (TNLs), CC (CNLs), or RPW8 (RNLs) genes. We manually examined these NLR genes to identify those with alternative polyadenylation. Reannotation of some loci was required to interpret the effects of FPA activity. For example, we found that the TNL gene *AT5G46490*, located in the RPS6 cluster, is incorrectly annotated as two loci, *AT5G46490* and *AT5G46500* (*Figure 3—figure supplement 1*). Nanopore DRS evidence indicates that this is actually a single locus with a previously unrecognised 2.7 kb intron containing a proximal poly(A) site, the use of which is controlled by FPA. This interpretation is supported by nanoPARE data (*Schon et al., 2018*), which showed no evidence of capped 5′ ends originating from the annotated downstream gene. Use of the distal poly(A) site introduces an additional ~400 amino acids to the C-terminus of the protein. This C-terminal region has homology to other NLRs in the *RPS6* cluster and is predicted to introduce additional LRR repeats (*Martin et al., 2020*; *Figure 3—figure supplement 2*).

Notably, we could also reannotate the chromosomal region around *RPS6* itself. The extreme autoimmunity phenotypes of NMD mutants and mitogen-activated kinase pathway mutants require

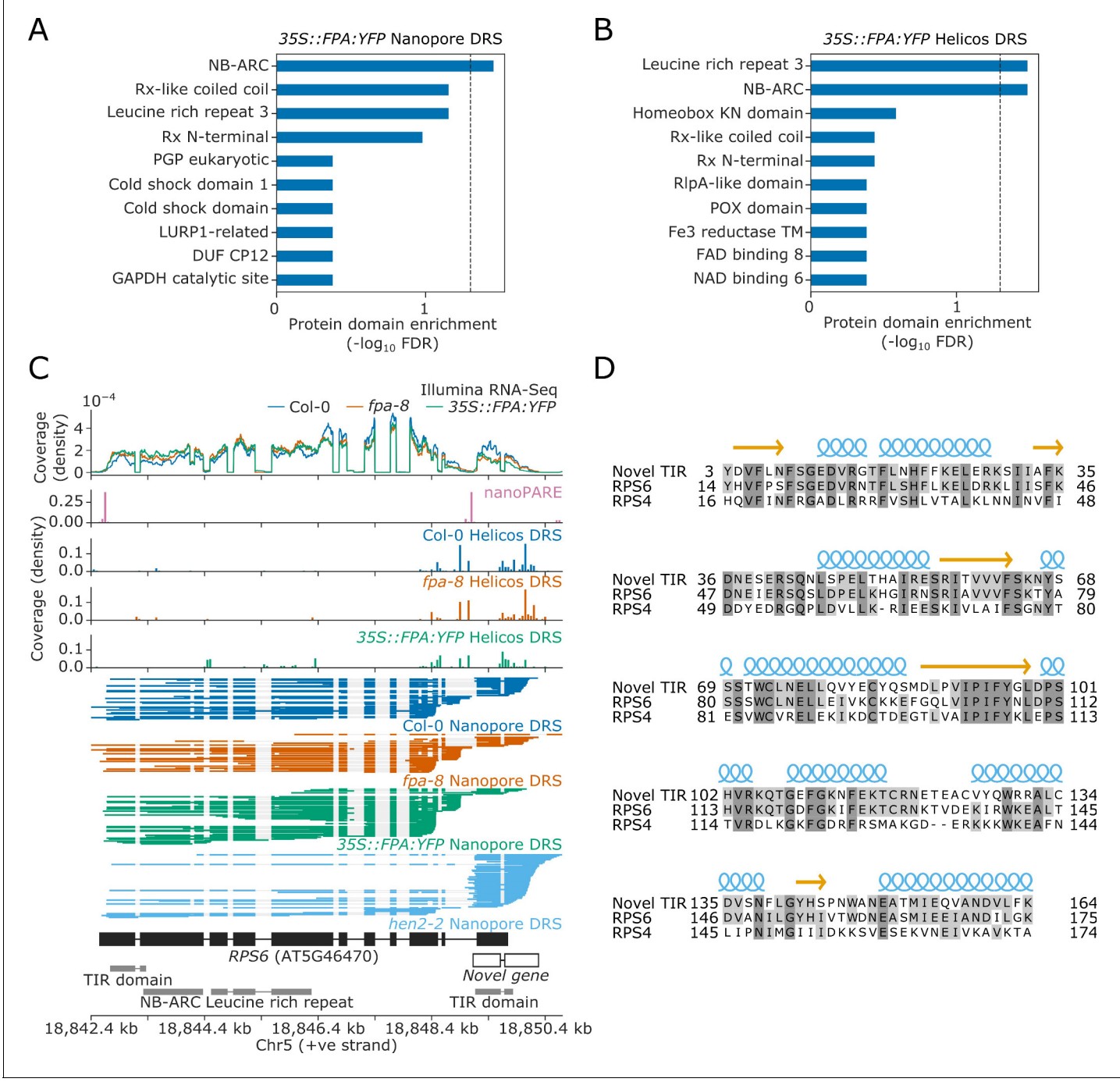

**Figure 3.** Nanopore and Helicos DRS identify NLR genes regulated by alternative polyadenylation. (**A–B**) Protein domain enrichment analysis for loci with increased proximal poly(A) site selection in *35S::FPA:YFP* line, as detected using (**A**) Nanopore DRS or (**B**) Helicos DRS. (**C**) Nanopore DRS reveals the complexity of RNA processing at *RPS6*. Protein domain locations (shown in grey) represent collapsed InterPro annotations. The novel TIR domain was annotated using InterProScan (***Mitchell et al., 2019***). (**D**) Protein alignment of the predicted TIR domain from the novel gene downstream of RPS6, with the sequence of the TIR domains from RPS6 and RPS4. Helix and strand secondary structures (from UniProt: RPS4, Q9XGM3) are shown in blue and yellow, respectively. Residues are shaded according to the degree of conservation.

The online version of this article includes the following figure supplement(s) for figure 3:

**Figure supplement 1.** Nanopore DRS informs reannotation of the complex NLR locus encompassing the *AT5G46490* and *AT5G46500* annotations.

**Figure supplement 2.** Nanopore DRS informs reannotation of the complex NLR locus encompassing the *AT5G46490* and *AT5G46500* annotations.

RPS6 but the mechanisms involved are not understood (*Gloggnitzer et al., 2014*; *Takagi et al., 2020*). Nanopore DRS indicates that the 3′UTR of *RPS6* is complex, with multiple splicing events and poly(A) sites (*Figure 3C*). We also detected transcripts expressed from this region that do not appear to be contiguous with *RPS6* 3′UTR reads. Instead, these reads correspond to an independent unannotated gene that overlaps the *RPS6* 3′UTR. This interpretation is supported by capped RNA 5′ ends detected in this region by nanoPARE (*Schon et al., 2018*). In addition, Nanopore DRS analysis of the RNA exosome mutant *hen2-2* (*Parker et al., 2021*) revealed that this unannotated gene is expressed at relatively high levels, but that the transcripts are subject to degradation. Consequently, steady-state levels of RNA expressed from this locus are relatively low in Col-0. The gene encodes a TIR domain similar to that of RPS6 (*Figure 3D*). Therefore, use of the distal *RPS6* poly(A) site constitutes readthrough into the downstream TIR-domain-only NLR. Based on these analyses, we conclude that long-read Nanopore DRS data have the potential to correct NLR gene annotation at complex loci that cannot be resolved by genome annotation software or short-read Illumina RNA-Seq.

## Widespread premature transcription termination of NLRs includes frequent selection of poly(A) sites in protein-coding exons

Of the 206 NLR genes examined, 124 had a sufficient level of expression to identify alternative polyadenylation in the Nanopore DRS data; of these 124, 62 (50.0%) were found to have FPA-dependent alternative polyadenylation (*Tables 1–3*). Of the 74 expressed NLRs located in major clusters, 44 (59.5%) were sensitive to FPA activity ($chi^2$p=0.02) (*Lee and Chae, 2020*). The localisation of NLRs to large genomic clusters is known to facilitate diversification (*Barragan and Weigel, 2020*). Consistent with this, 20 (71.4%) of the 28 expressed NLRs reported to be under high levels of diversifying selection were sensitive to FPA activity ($chi^2$p=0.02) (*Prigozhin and Krasileva, 2021*). In addition, FPA-sensitive NLRs tended to be located in regions with higher levels of synteny diversity (*Jiao and Schneeberger, 2020*), although in this case the association was not significant (*t*-test p=0.09; *Figure 4—figure supplement 1*). Overall, these findings suggest that FPA-dependent alternative polyadenylation is associated with rapidly evolving NLRs.

The effects of FPA activity can be broadly classified into three modes of control involving (i) readthrough and chimeric RNAs, (ii) intronic poly(A) sites, and (iii) poly(A) sites within protein-coding exons. At certain complex loci, FPA can affect poly(A) site choice using combinations of these different modes of regulation.

**Table 1.** Readthrough and chimeric RNA formation events at FPA-sensitive NLR genes.

| Gene ID | Gene name | NLR class | Chimeric pair (upstream–downstream) |
|---|---|---|---|
| AT1G12220 | RPS5 | CNL | AT1G12220–AT1G12230 |
| AT1G58848 | RPP7a/b | TNL | AT1G58848–AT1G58889 |
| AT1G59218 | RPP7a/b | TNL | AT1G59218–AT1G59265 |
| AT1G61190 | - | CNL | ncRNA–AT1G61190 |
| AT1G63730 | - | TNL | AT1G63730–AT1G63740 |
| AT1G63740 | - | TNL | AT1G63730–AT1G63740 |
| AT3G46730 | - | CNL | AT3G46740–AT3G46730 |
| AT4G16860 | RPP4 | TNL | AT4G16860–AT4G16870–AT4G16857 |
| AT4G16960 | SIKIC3 | TNL | AT4G16970–AT4G16960–AT4G16957 |
| AT4G19060 | - | NB only | AT4G19070–AT4G19060 |
| AT4G19530 | - | TNL | AT4G19530–AT4G19540 |
| AT5G38850 | - | TNL | AT5G38850–AT5G38860 |
| AT5G40090 | CHL1 | TNL | ncRNA–AT5G40090 |
| AT5G44510 | TAO1 | TNL | AT5G44520–AT5G44510 |
| AT5G45490 | - | CNL | AT5G45472–AT5G45490 |
| AT5G46470 | RPS6 | TNL | AT5G46470–TIR gene |
| AT5G48780 | - | TNL | AT5G48775–AT5G48780 |

**Table 2.** Intronic proximal polyadenylation events at FPA-sensitive NLR genes.

| Gene ID | Gene name | NLR class | Predicted function | Protein isoform |
|---|---|---|---|---|
| AT1G12210 | RFL1 | CNL | non-coding (5'UTR) | - |
| AT1G58602 | RPP7 | CNL | non-coding (5'UTR); alternative 3'UTR | - |
| AT1G63750 | WRR9 | TNL | protein coding | TIR only |
| AT1G63880 | RLM1B | TNL | protein coding; non-stop | TIR only |
| AT1G69550 | - | TNL | protein coding | LRR truncation |
| AT3G44480 | RPP1 | TNL | protein coding | LRR truncation |
| AT3G50480 | HR4 | RPW8 | protein coding | RPW8 truncation |
| AT4G16860 | RPP4 | TNL | protein coding | TIR only |
| AT4G16900 | - | TNL | protein coding | LRR truncation |
| AT4G19510 | RPP2B | TNL | alternative 3'UTR | - |
| AT5G17890 | DAR4/CHS3 | TNL | protein coding | TIR only |
| AT5G40910 | - | TNL | protein coding | TIR only |
| AT5G43730 | RSG2 | CNL | non-coding (5'UTR) | - |
| AT5G43740 | - | CNL | non-coding (5'UTR) | - |
| AT5G46270 | - | TNL | protein coding | TIR/NB-ARC only; LRR truncation |
| AT5G46470 | RPS6 | TNL | alternative 3'UTR | - |
| AT5G46490 | - | TNL | protein coding; non-stop | TIR/NB-ARC only; LRR truncation |

For 17 NLR genes, we found that a change in FPA activity altered the formation of readthrough or chimeric RNAs containing one or more NLR loci (*Table 1*). The duplicated *RPP7a/b*-like genes *AT1G58848* and *AT1G59218* (which form part of the *RPP7* cluster containing five CNL-class NLRs) displayed increased readthrough into downstream transposable elements (TEs) in *fpa-8* (*Figure 4A*). EMD tests could not be performed at these loci due to the multi-mapping of reads at these duplicated genes (*AT1G58848* and *AT1G59218*). Loss of FPA function can also lead to clusters of two or more NLR genes being co-transcribed as a single transcriptional unit. For example, the TNL-class gene *AT1G63730*, located in the *B4/RLM1* cluster, forms chimeric RNA with the downstream TNL-class gene *AT1G63740* in *fpa-8* (Helicos EMD = 1099, FDR = 0.02; *Figure 4—figure supplement 2*).

We identified another 17 NLR genes with intronic polyadenylation controlled by FPA (*Table 2*). Of these, four contained poly(A) sites in 5'UTR introns (which would result in non-coding transcripts) and three contained alternative poly(A) sites after the stop codon (which could alter potential regulatory sequences contained in 3'UTRs). The remainder contained poly(A) sites in introns between protein-coding exons. Selection of these poly(A) sites introduce premature stop codons that result in truncated open reading frames (ORFs). For example, we identified a proximal poly(A) site within the third intron of *AT1G69550*, which encodes a TNL-type singleton NLR (*Figure 4B*). Use of this poly(A) site results in mRNAs with a premature stop codon; the encoded protein lacks most of the predicted LRR domain. In *fpa-8*, readthrough at this poly(A) site is increased (Helicos EMD = 1271, FDR = $1.2 \times 10^{-4}$), resulting in an increase in the relative number of full-length transcripts.

The most common form of FPA-dependent NLR regulation was premature termination within exons (*Table 3*). We identified 45 NLRs controlled in this way: at 44 of these loci, termination occurred within protein-coding exons. In most cases, this results in stop-codonless transcripts that are predicted targets of non-stop decay (*Szádeczky-Kardoss et al., 2018*). Many of these proximal exonic poly(A) sites could be identified at lower levels in Col-0. For example, at *RPP28* (AT2G14080), which encodes a TNL-class singleton NLR, we detected multiple exonic poly(A) sites located within the second and fourth exons, which encode the NB-ARC and LRR domains, respectively (*Figure 4C*). Selection of these exonic poly(A) sites was increased in *35S::FPA:YFP* (Helicos EMD = 859, FDR = $5.4 \times 10^{-9}$) and decreased in *fpa-8* (Helicos EMD = 912, FDR = $7.6 \times 10^{-9}$). FPA

**Table 3.** Exonic proximal polyadenylation events at FPA-sensitive NLR genes.

| Gene ID | Gene name | NLR class | Predicted function | Protein isoform |
|---|---|---|---|---|
| AT1G10920 | LOV1 | CNL | protein coding* | CC-only* |
| AT1G27180 | - | TNL | non-stop | - |
| AT1G31540 | RAC1 | TNL | non-stop; protein coding^ | LRR truncation^ |
| AT1G33560 | ADR1 | RNL | non-stop | - |
| AT1G53350 | - | CNL | non-stop | - |
| AT1G56510 | WRR4A | TNL | non-stop | - |
| AT1G56520 | - | TNL | non-stop | - |
| AT1G58602 | RPP7 | CNL | non-stop | - |
| AT1G58807 | RF45 | CNL | non-stop | - |
| AT1G58848 | RPP7a/b | CNL | non-stop | - |
| AT1G59124 | RDL5 | CNL | non-stop | - |
| AT1G59218 | RPP7a/b | CNL | non-stop | - |
| AT1G61300 | - | CNL | non-stop | - |
| AT1G62630 | - | CNL | non-stop | - |
| AT1G63360 | - | CNL | non-stop | - |
| AT1G63730 | - | TNL | non-stop | - |
| AT1G63860 | - | TNL | non-stop | - |
| AT1G63880 | RLM1B | TNL | non-stop | - |
| AT1G72840 | - | TNL | non-coding (5'UTR) | - |
| AT2G14080 | RPP28 | TNL | non-stop | - |
| AT3G44480 | RPP1 | TNL | non-stop; protein coding[†] | LRR truncation[†] |
| AT3G44630 | - | TNL | non-stop | - |
| AT3G44670 | - | TNL | non-stop; protein coding[†] | TIR only[†] |
| AT3G46530 | RPP13 | CNL | non-stop | - |
| AT4G16860 | RPP4 | TNL | non-stop | - |
| AT4G16890 | SNC1 | TNL | non-stop | - |
| AT4G16900 | - | TNL | non-stop | - |
| AT4G19520 | - | TNL | non-stop | - |
| AT4G19530 | - | TNL | non-stop | - |
| AT4G36140 | - | TNL | non-stop | - |
| AT5G17890 | DAR4/CHS3 | TNL | non-stop | - |
| AT5G35450 | - | CNL | non-stop | - |
| AT5G38850 | - | TNL | non-stop | - |
| AT5G40060 | - | TNL | protein coding* | TIR only* |
| AT5G40910 | - | TNL | non-stop | - |
| AT5G43470 | RPP8 | CNL | non-stop | - |
| AT5G43740 | - | CNL | non-stop | - |
| AT5G44510 | TAO1 | TNL | non-stop; protein coding[†] | LRR truncation[†] |
| AT5G44870 | LAZ5 | TNL | non-stop | - |
| AT5G45050 | RRS1B | TNL | non-stop | - |
| AT5G45250 | RPS4 | TNL | protein coding[†] | LRR truncation[†] |
| AT5G45260 | RRS1 | TNL | non-stop | - |
| AT5G46270 | - | TNL | non-stop; protein coding[†] | LRR truncation[†] |
| AT5G48620 | - | CNL | non-stop | - |
| AT5G58120 | DM10 | TNL | non-stop; protein coding[†] | LRR truncation[†] |

* indicates loci where exonic proximal polyadenylation generates transcripts that may be protein coding due to upstream ORFs.

† indicates loci where exonic proximal polyadenylation coupled with intron retention results in a protein-coding ORF.

was also found to promote premature termination in the protein-coding sequence of single-exon, intronless NLR genes. For example, at *RPP13* (AT3G46530), which encodes a CNL-class NLR protein, FPA overexpression causes selection of proximal poly(A) sites located within the region encoding the LRR domain (Helicos EMD = 228, FDR = $1.8 \times 10^{-4}$; *Figure 4—figure supplement 3*).

Although the most frequent consequence of FPA selection of exonic poly(A) sites was stop-codonless transcripts, we also identified examples where the protein-coding potential was altered. For example, *AT5G40060* encodes a TNL-class NLR but has a premature stop codon between the TIR and NB-ARC domains. Consequently, full-length transcription results in an mRNA with an upstream ORF (uORF) encoding the TIR domain and a larger downstream ORF encoding NB-ARC and LRR domains (*Figure 4D*). However, transcripts with such large uORFs are targets of NMD in plants (*Nyikó et al., 2009*). Therefore, FPA-dependent proximal polyadenylation in the region encoding the NB-ARC domain results in a transcript containing only the uORF, which is not a predicted NMD target and may be more efficiently translated into a TIR-only protein.

In seven of the identified genes, exonic proximal polyadenylation is associated with retention of an upstream intron (*Table 3*). As a result, premature stop codons are introduced, resulting in a truncated coding region. For example, the TNL-type NLR *RPS4* was previously shown to be regulated by alternative splicing induced by the effector AvrRps4 (*Zhang and Gassmann, 2007*). We identified an increase in *RPS4* intron 3 retention in *35S::FPA:YFP* compared with Col-0 that was associated with proximal polyadenylation events in exon 4 (Helicos EMD = 34, not significant; *Figure 4—figure supplement 4*). Therefore, inter-dependence between splicing and poly(A) site choice may explain *RPS4* control.

FPA controlled NLR poly(A) site selection at 16 complex loci with combinations of intronic, exonic, and readthrough sites. One example is *RPP4* (AT4G16860), a TNL-class NLR known to mediate Arabidopsis resistance to *Hpa* isolate Emoy2 (*Hpa*-Emoy2) (*van der Biezen et al., 2002*). *RPP4* is part of the *RPP5* cluster, comprising seven TNL-class NLRs. In agreement with a previous study (*Wang and Warren, 2010*), we found that in wild-type Col-0, *RPP4* can be transcribed as a chimeric RNA together with the downstream *AtCOPIA4* TE (AT4G16870) through selection of one of the two distal poly(A) sites located within the TE (*Figure 5*; *Wang and Warren, 2010*) or selection of a third poly(A) site in the downstream gene *AT4G16857*. Use of the proximal poly(A) site within the TE is associated with an approximately 8 kb cryptic splicing event between the 5′ splice site of the first exon of *RPP4* and a 3′ splice site located within the TE. Both Nanopore DRS and Illumina RNA-Seq data provide evidence for this cryptic splicing event, which skips all *RPP4* exons downstream of exon 1, removing most of the *RPP4* coding sequence and introducing a stop codon (*Figure 5*, Inset 1). The resulting transcript is predicted to encode a TIR-domain-only protein. Loss of FPA function decreases chimeric RNA formation by shifting poly(A) site selection towards a proximal poly(A) site located within the protein-coding region of the final exon (*Figure 5—figure supplement 1*). This results in the production of *RPP4* transcripts lacking in-frame stop codons (*Figure 5*, Inset 2). Furthermore, in *35S::FPA:YFP*, we observed increased selection of a proximal poly(A) site located within the first intron of *RPP4*, which would also encode a truncated RPP4 protein. We conclude that FPA-dependent alternative polyadenylation at *RPP4* produces transcripts with unusually long 3′UTRs, alternative protein isoforms and transcripts that cannot be efficiently translated.

## FPA controls *RPP7* by promoting premature termination within protein-coding exon 6

To examine the functional impact of FPA on the regulation of NLRs, we focused on *RPP7*. *RPP7* encodes a CNL-class NLR protein which is necessary for resistance to *Hpa* isolate Hiks1 (*Hpa*-Hiks1) in Col-0 (*McDowell et al., 2000*). The full-length expression of *RPP7* is controlled by elongation factors that interact with H3K9me[2], which is associated with the COPIA-type retrotransposon (*COPIA-R7*) located in *RPP7* intron 1 (*Saze et al., 2013*). Using Nanopore and Helicos DRS data, we identified at least two poly(A) sites within the *COPIA-R7* element, both of which were selected more frequently in

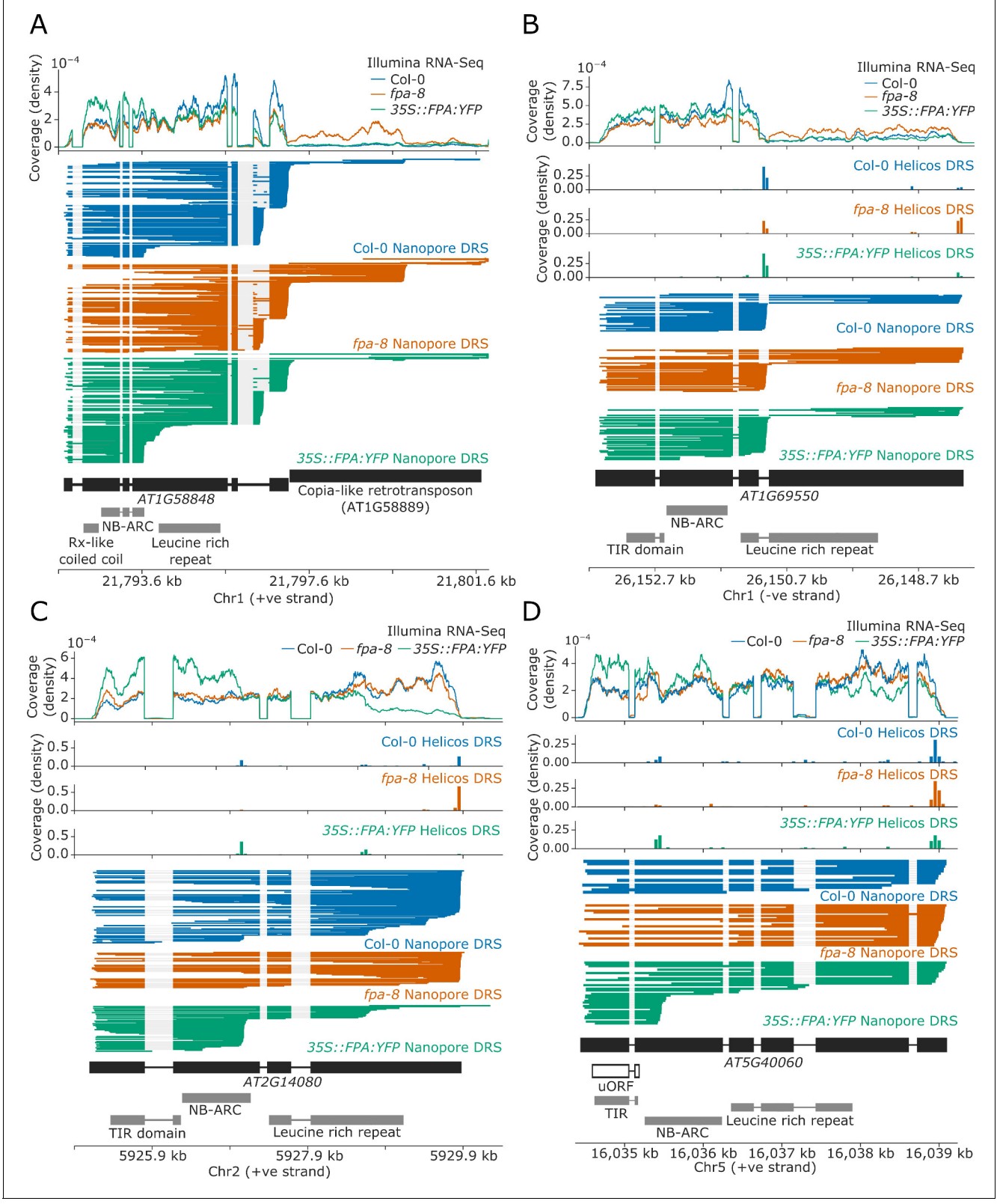

**Figure 4.** FPA-dependent alternative polyadenylation of NLR transcripts. FPA controls (**A**) readthrough and chimeric RNA formation at *AT1G58848* (unique mapping of short Helicos DRS reads was not possible due to the high homology of *AT1G58848* to tandemly duplicated NLR loci in the same cluster); (**B**) intronic polyadenylation at *AT1G69550*, resulting in transcripts encoding a protein with a truncated LRR domain; (**C**) exonic polyadenylation

*Figure 4 continued on next page*

*Figure 4 continued*

at *AT2G14080*, resulting in stop-codonless transcripts; and (D) exonic polyadenylation at *AT5G40060*, resulting in transcripts encoding a TIR-domain-only protein due to an upstream ORF.

The online version of this article includes the following figure supplement(s) for figure 4:

**Figure supplement 1.** NLR genes with FPA-dependent alternative polyadenylation are found in hotspots of rearrangements.
**Figure supplement 2.** Loss of FPA function causes chimeric RNA formation at *AT1G63730* and *AT1G63740* NLR loci.
**Figure supplement 3.** FPA overexpression increases exonic proximal polyadenylation of *RPP13*.
**Figure supplement 4.** FPA overexpression causes intron retention and exonic proximal polyadenylation at intron 3 of *RPS4*.

*fpa-8* (*Figure 6A*, *Figure 6—figure supplement 1*). We also identified two poly(A) sites within the second intron of *RPP7*. The use of both sites is reciprocally sensitive to FPA activity, with a moderate decrease in *fpa-8* and an increase in *35S::FPA:YFP*. All these intronic proximal poly(A) sites are located before the start of the *RPP7* ORF and generate transcripts that do not encode RPP7 protein. At the 3′ end of *RPP7*, we found three alternative poly(A) sites located in the terminal intron, in addition to the previously reported most distal and most commonly used poly(A) site in the terminal exon (*Figure 6A*, Inset 1) (*Tsuchiya and Eulgem, 2013*). Selection of each of these poly(A) sites is associated with alternative splicing events that lead to the generation of four possible 3′UTR sequences. Termination at the 3′UTR intronic poly(A) sites is suppressed by FPA: their usage is increased in *fpa-8* and decreased in *35S::FPA:YFP*. These data indicate that FPA influences *RPP7* intronic polyadenylation at a larger number of poly(A) sites than previously supposed.

The major effect of FPA on *RPP7* is within protein-coding exon 6, where we identified three poly(A) sites (*Figure 6A*, Inset 2): two at the end of the region encoding the NB-ARC domain and one within the region encoding the LRR repeats. Cleavage and polyadenylation at these sites result in transcripts without in-frame stop codons, thereby disrupting the coding potential of *RPP7* mRNA. These poly(A) sites were identified in both Helicos and Nanopore DRS data, indicating that they are unlikely to be caused by alignment errors. The relative selection of exon 6 poly(A) sites depends on FPA expression: in Col-0, 25% of *RPP7* Nanopore DRS reads terminate at one of these exon 6 poly(A) sites; and when FPA is overexpressed, this figure increases to 63%. Consistent with this, a relative drop in coverage at exon 6 was also observed in *35S::FPA:YFP* Illumina RNA-Seq data. Consequently, only 23% of *RPP7* transcripts are expected to encode RPP7 protein in the FPA-overexpressing line. In contrast, 4% of *RPP7* Nanopore DRS reads identified in *fpa-8* terminate in exon 6, and 79% of transcripts are expected to be protein coding. In an orthogonal approach, we used RNA gel blot analysis to visualise *RPP7* mRNAs in Col-0, *fpa-8*, and *35S::FPA:YFP* backgrounds and detected a clear decrease in signal corresponding to full-length *RPP7* transcripts in *35S::FPA:YFP* (*Figure 6B*). These data support previous evidence of FPA-dependent control of *RPP7* (*Deremetz et al., 2019*) but reveal that the predominant mechanism is via exonic transcription termination.

## *RPP7*-dependent immunity to the biotrophic pathogen *Hpa* is sensitive to FPA expression

We next asked whether FPA-dependent premature transcription termination at *RPP7* exon 6 has a functional consequence. Since FPA reduced the level of full-length protein-coding *RPP7* transcripts, we asked whether increased FPA activity might compromise RPP7-dependent immunity. To test this hypothesis, we carried out pathogenesis assays using the oomycete strain *Hpa*-Hiks1. RPP7 function is required for immunity to *Hpa*-Hiks1 in Col-0 (*McDowell et al., 2000*). The Keswick (Ksk-1) accession is susceptible to *Hpa*-Hiks1 (*Lai et al., 2019*) and we used it as a control in these studies.

We inoculated Arabidopsis seedlings with *Hpa*-Hiks1 spores in three independent experiments. Four days after inoculation, we checked susceptibility by counting the number of sporangiophores. With the exception of Ksk-1, all of the lines we tested were in a Col-0 background. As expected, Col-0 plants were resistant to infection (median: 0 sporangiophores per plant), and Ksk-1 plants were sensitive to infection (median: five sporangiophores per plant; $p=1.7 \times 10^{-32}$; *Figure 6C*). *fpa-7* mutants were as resistant to infection as Col-0 (median: 0 sporangiophores per plant, p=0.19). This is consistent with our finding that full-length *RPP7* transcript expression is not reduced in the absence of FPA. *fpa-8* mutants were also resistant to infection (median: 0 sporangiophores per plant); however, there was slight variability in their resistance compared to *fpa-7* ($p=2.4 \times 10^{-12}$).

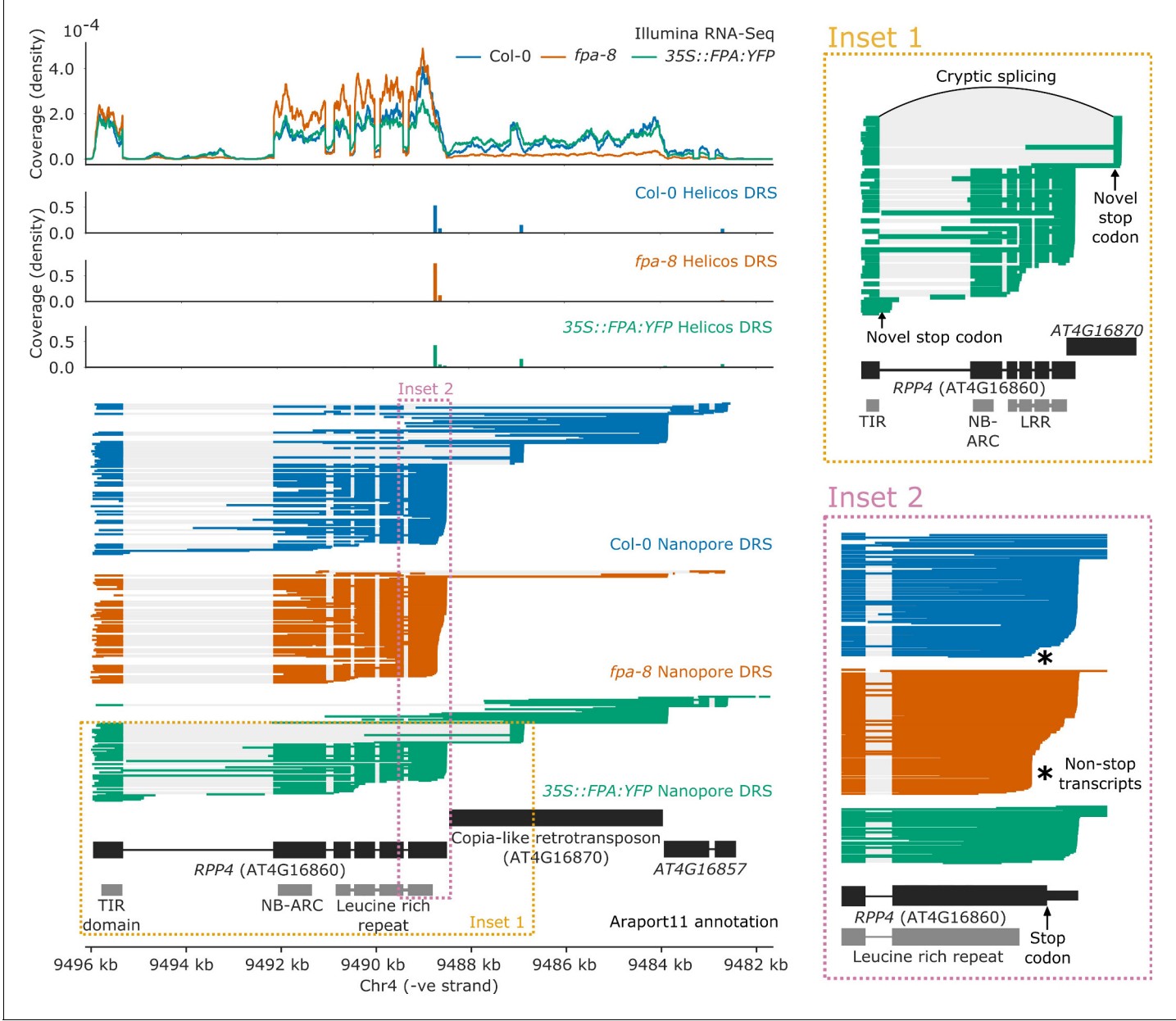

**Figure 5.** Complex FPA-dependent patterns of alternative polyadenylation at *RPP4*. FPA-dependent intronic, exonic and readthrough poly(A) site selection in *RPP4*. (Inset 1) A magnified view of TIR-domain-only *RPP4* transcripts detected in *35S::FPA:YFP* caused by proximal polyadenylation in intron 1, and distal polyadenylation within the TE associated with cryptic splicing. (Inset 2) A magnified view of the stop-codonless transcripts produced within the protein-coding *RPP4* region in *fpa-8*.

The online version of this article includes the following figure supplement(s) for figure 5:

**Figure supplement 1.** Complex FPA-dependent patterns of alternative polyadenylation at the *RPP4* locus.

This variability was not restored by complementation with a *pFPA::FPA* transgene (p=0.23) indicating that it is not caused by loss of FPA function, and is likely to result from other mutations in the *fpa-8* background. In contrast, *35S::FPA:YFP* plants were significantly more sensitive to *Hpa*-Hiks1 than *pFPA::FPA* (median: three sporangiophores per plant; p=3.8 × 10$^{-9}$), indicating that overexpression of FPA compromises immunity. We conclude that FPA control of poly(A) site selection can modulate NLR function, with a functional consequence for immunity.

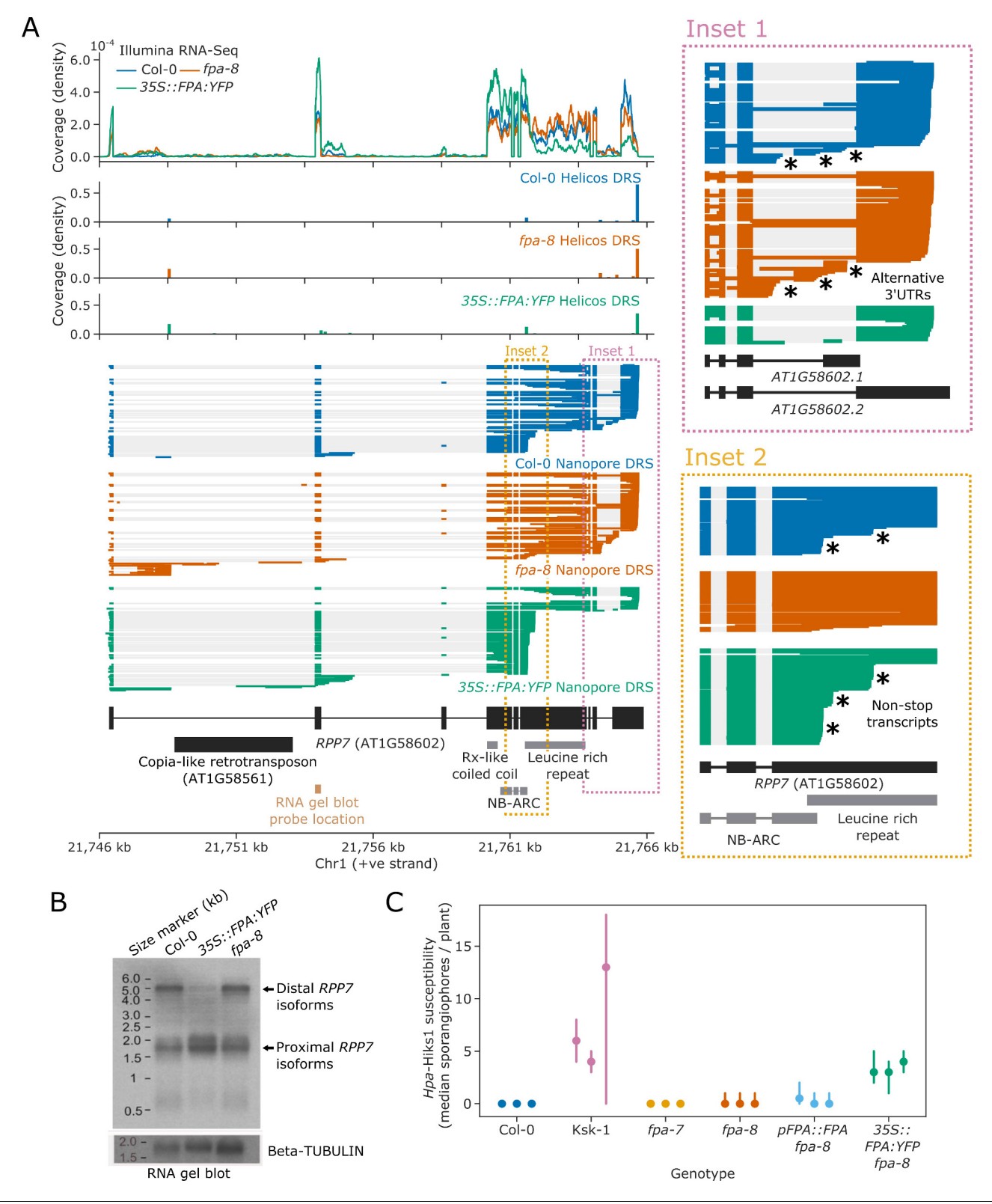

**Figure 6.** FPA promotes premature cleavage and polyadenylation within *RPP7* protein-coding exon six that compromises plant immunity against *Hpa*-Hiks1. (**A**) FPA-dependent RNA 3′ end formation changes at the *RPP7 (AT1G58602)* locus. (Inset 1) Magnified view of the *RPP7* 3′UTR region with alternative RNA 3′ ends. (Inset 2) Magnified view of the stop-codonless transcripts produced in protein-coding *RPP7* exon 6. (**B**) RNA gel blot visualising *RPP7* transcripts in Col-0, *fpa-8* and *35S::FPA:YFP*. Probe location in second exon is shown on (**A**) (light brown). Beta-TUBULIN was used as an internal

*Figure 6 continued on next page*

*Figure 6 continued*

control. (C) FPA-dependent premature exonic termination of *RPP7* compromises immunity against *Hpa*-Hiks1. Point plot showing median number of sporangiophores per plant calculated 4 days after *Hpa*-Hiks1 inoculation. Error bars are 95% confidence intervals. Each experimental replicate was generated from 7 to 45 plants per genotype.

The online version of this article includes the following source data and figure supplement(s) for figure 6:

**Source data 1.** *Hpa*-Hiks1 susceptibility results for the Col-0, Ksk-1, *fpa-7*, *fpa-8*, *pFPA::FPA* and *35S::FPA:YFP* lines [Linked to *Figure 6C*].

**Figure supplement 1.** Complex FPA-dependent patterns of alternative polyadenylation at the *RPP7* locus.

## Discussion

We have identified a novel role for the RNA-binding protein FPA in the control of plant innate immunity. Using *IVI-MS* proteomics and ChIP-Seq, we showed that FPA is closely associated with proteins involved in RNA 3′ processing and co-localises with Ser$^2$ phosphorylated Pol II at the 3′ ends of genes. Integrative analysis using three RNA sequencing technologies confirmed that the major effect of modulating FPA activity is to alter poly(A) site selection. An unexpected finding was that half of expressed NLR loci were sensitive to FPA activity. In most cases, FPA promoted the use of poly(A) sites within protein-coding exons of NLR genes. At *RPP7*, an increase in exonic polyadenylation caused by FPA overexpression was shown to compromise immunity to *Hpa*-Hiks1. The widespread nature of this control mechanism suggests that transcription termination plays an important role in the regulatory and evolutionary dynamics of NLR genes.

### Uncovering protein assemblies that mediate 3′ end processing in living plant cells

We used an in vivo formaldehyde cross-linking approach to identify proteins that co-localise with FPA inside living plant cells. These data provide in-depth knowledge of the proteins involved in Arabidopsis RNA 3′ end processing and clues to the function of the uncharacterised proteins identified here. Components of the m$^6$A writer complex also co-purify with FPA. However, unlike related proteins in human and *Drosophila* (*Knuckles et al., 2018*; *Patil et al., 2016*), we found that FPA is not required to maintain global levels of m$^6$A modification in Arabidopsis.

Two Arabidopsis PCF11 paralogs with Pol II CTD-interacting domains (CIDs), PCFS2 and PCFS4, co-purified with FPA, but two paralogs lacking CIDs, PCFS1, and PCFS5, did not. PCF11 was previously shown to have functionally separable roles in transcription termination and cleavage and polyadenylation (*Sadowski et al., 2003*): the N-terminal PCF11 CID is required for transcription termination, whereas the C-terminal domains are required for cleavage and polyadenylation. The specific interaction of FPA with CID-containing PCF11 paralogs suggests that FPA controls alternative polyadenylation by altering Pol II speed and transcription termination. The human SPOC domain protein PHF3 can bind to two adjacent Ser$^2$ phosphorylated heptads of the CTD of Pol II via two electropositive patches on the surface of its SPOC domain (*Appel et al., 2020*). One of these patches, and the key amino acid residues within it, is conserved in the structure of the FPA SPOC domain (*Zhang et al., 2016*). Consequently, FPA might also interact with the CTD, possibly in conjunction with CID domains of PCFS2 and PCFS4. Such interactions could account for the global correlation between FPA and Pol II Ser$^2$ occupancy and explain how FPA is able to associate with terminating Pol II at the 3′ ends of most expressed genes.

### Widespread control of NLR transcription termination by FPA

An unanticipated finding of this study is that Arabidopsis NLR genes were enriched amongst loci with FPA-sensitive poly(A) sites. NLRs function in the immune response and, consistent with this crucial role, they are under powerful and dynamic selective pressure. Defining the inventory of Arabidopsis NLRs depended on long-range DNA sequencing of diverse accessions (*Van de Weyer et al., 2019*). Here, we show that long-read Nanopore DRS provides insight into the authentic complexity of NLR mRNA processing and enables the accurate annotation of NLR genes. For example, our reannotation of the *RPS6* locus is essential to understand the recurring role of RPS6 in autoimmunity. The autoimmune phenotypes of mutants defective in NMD or the mitogen-activated kinase pathway are

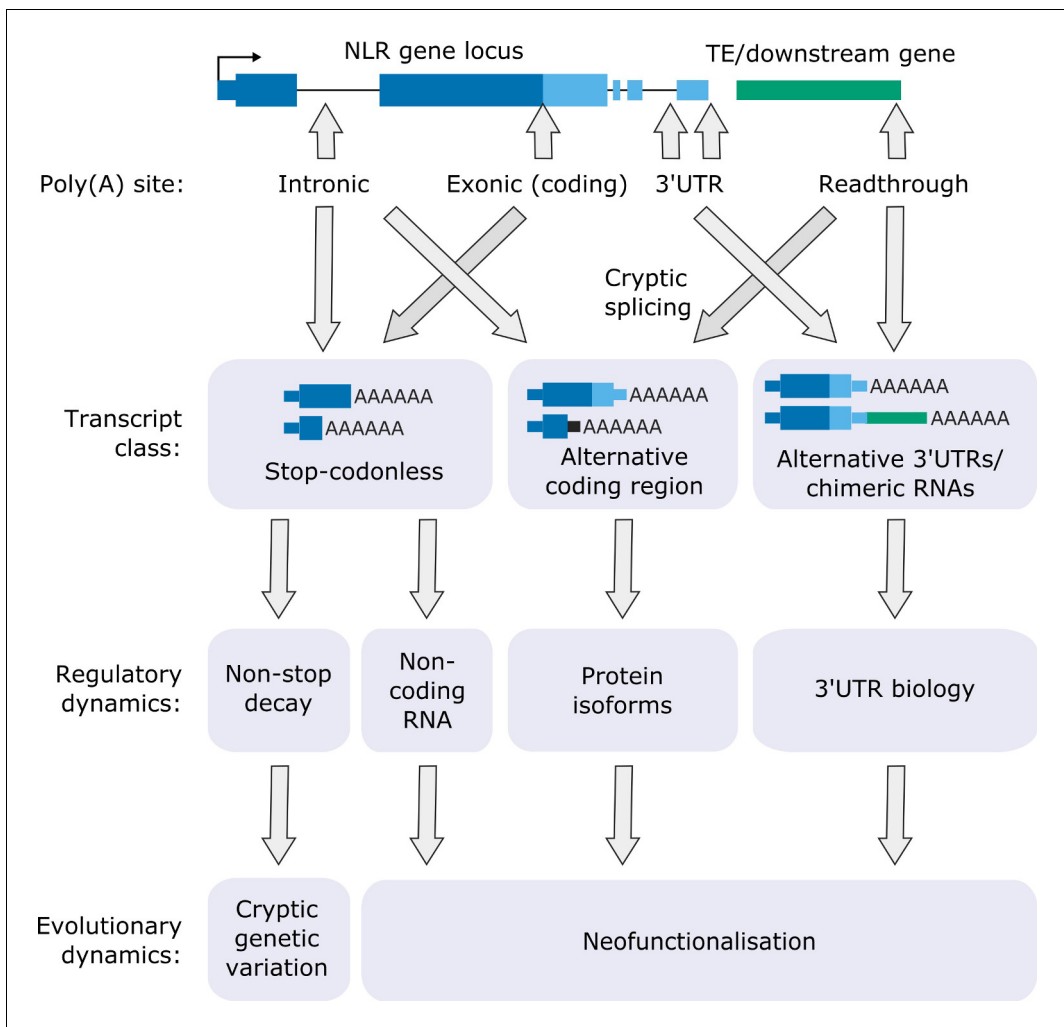

**Figure 7.** Functional consequences of FPA-dependent alternative polyadenylation at NLR loci. Model diagram showing how FPA-dependent alternative polyadenylation at NLR loci might affect the regulatory and evolutionary dynamics of plant disease resistance.

RPS6 dependent, but the mechanisms involved are unclear (*Gloggnitzer et al., 2014*; *Takagi et al., 2020*). We found that *RPS6* is transcribed through a previously unrecognised downstream gene that encodes an RPS6-like TIR domain. We showed that expression of the downstream gene is dependent on the RNA exosome component HEN2. In addition, mutations in *HEN2* were recently identified as suppressors of RPS6-dependent autoimmune phenotypes (*Takagi et al., 2020*). It is clear that accurate annotation of complex NLR loci facilitates the interpretation of basic features of NLR function.

Of the 124 NLRs with detectable expression in Nanopore DRS data, 62 were sensitive to FPA activity. FPA controls 3′ end formation of NLR genes in three different transcript locations (*Figure 7*): (i) 3′UTRs, where it can prevent readthrough and chimeric RNA formation; (ii) introns, where it promotes proximal polyadenylation; and (iii) protein-coding exons, where it promotes stop-codonless transcript formation. The consequences of such complex control of RNA 3′ end formation are wide-ranging and likely to be context dependent (*Mayr, 2019*).

Where FPA controls readthrough and chimeric RNA formation, it affects 3′UTR length, sequence composition and cryptic splice site usage. Long or intron-containing 3′UTRs are targeted by NMD, leading to RNA decay or suppressed translatability. Long, unstructured 3′UTRs influence

intermolecular RNA interactions and phase separation, changing the subcellular localisation of mRNAs (*Ma et al., 2020*). The close proximity of mRNAs in the resulting granules may enable co-translational protein complex formation. Readthrough transcription may also disrupt the expression of downstream genes by transcription interference (*Proudfoot, 1986*).

FPA-dependent premature transcription termination at intronic poly(A) sites can introduce novel stop codons, resulting in transcripts that encode truncated NLR proteins with altered functions. For example, some TIR domain-only proteins are known to be active in NLR regulation, resulting in constitutive signalling activity (*Zhang et al., 2004*) or act as competitive inhibitors by titrating full-length NLR protein (*Williams et al., 2014*). In other cases, TIR-domain-only proteins are sufficient for pathogen recognition (*Nishimura et al., 2017*). The TE-containing 3′UTR of *RPP4* appears to be required for resistance to the pathogen *Hpa*-Emoy2, although the mechanism involved is unclear (*Wang and Warren, 2010*). We discovered that cryptic splicing of *RPP4* exon 1 to a novel 3′ splice site within the TE can produce a unique transcript that encodes only the RPP4 TIR domain. It will be interesting to examine whether the TIR-only RPP4 isoform is required for full pathogen resistance. We also found that intron retention at *RPS4,* which is essential for RPS4-dependent resistance against *P. syringae* DC3000 (*Zhang and Gassmann, 2007*), is linked to exonic proximal polyadenylation. Intron retention without accompanying proximal polyadenylation will result in transcripts with long 3′UTRs that are likely to be sensitive to NMD, whereas proximally polyadenylated transcripts could be translated into truncated protein. Therefore, a combination of alternative polyadenylation and splicing probably underpins *RPS4* control. In future, sensitive proteomic analyses will be important to determine the impact of alternative polyadenylation on NLR protein isoform expression.

A remarkable finding was that FPA mostly targets the protein-coding exons of NLR genes and, even controls premature transcription termination within the ORF of single-exon NLR genes such as *RPP13*. Premature transcription termination in protein-coding exons results in the production of stop-codonless transcripts that cannot be efficiently translated into protein. These truncated transcripts may be subject to decay by RNA surveillance mechanisms (e.g. the non-stop decay pathway) or act as non-coding RNA decoys to titrate the levels of regulatory microRNAs (*Shivaprasad et al., 2012*). Increased rates of NLR transcription in plants under pathogen attack could promote elongation through such 'regulatory' poly(A) sites. In this way, the expression of NLR proteins might be restricted during pathogen surveillance but kept poised for rapid activation during infection.

Since the evolution of *cis*-regulatory elements controlling poly(A) site choice within introns or 3′UTRs is free from the constraints of protein-coding functionality, why should protein-coding exons be targeted so frequently? One possibility is that this enables the expression of newly created NLR genes to be kept under tight control, thereby facilitating rapid evolution whilst reducing the chances of autoimmunity (*Figure 7*). This hypothesis is strengthened by the finding that many NLRs with high allelic diversity (*Prigozhin and Krasileva, 2021*) are sensitive to FPA activity. Alternative polyadenylation might also function to hide NLR genes from negative selection and contribute to cryptic genetic variation in a similar way to the mechanism proposed for NMD- and microRNA-mediated NLR control (*Raxwal and Riha, 2016*; *Shivaprasad et al., 2012*). Cryptically spliced chimeric RNAs, with subsequent retrotransposition, can be a source of new genes (*Akiva et al., 2006*). Therefore, the control of transcription termination could directly facilitate the neofunctionalisation of NLRs. In the future, it will be important to compare patterns of transcription termination at NLRs across Arabidopsis accessions. For example, analysis of transcriptomic data will determine whether proximal polyadenylation is conserved in NLRs with high allelic diversity, whilst an integrative analysis of transcriptomic and genomic data will establish whether chimeric NLR transcripts identified in some accessions are found as retrotransposed genes in others.

At least two distinct patterns of alternative polyadenylation mediate *RPP7* regulation, one involving intronic heterochromatin (*Tsuchiya and Eulgem, 2013*) and another involving FPA-dependent termination in exon 6. The latter mechanism is conserved across all NLR genes of the Col-0 *RPP7* locus (*Table 3*). Alleles of these RPP7-like NLR genes have been identified as the causes of specific cases of hybrid necrosis (*Barragan et al., 2019*; *Chae et al., 2014*; *Li et al., 2020*). In these cases, autoimmunity is explained by allele-specific physical interactions between RPP7 protein and the RPW8-only protein HR4 (*Li et al., 2020*). We found that not only are *RPP7*-like genes targeted by FPA-dependent premature transcript termination, but so too is *HR4* (*Table 2*). This raises the possibility that FPA could rescue hybrid necrosis by limiting the expression of these proteins. FPA also appears to control the proximal polyadenylation of *DANGEROUS MIX 10 (DM10)*, producing

transcripts that could encode a protein with truncated LRR repeats. *DM10* alleles with LRR truncations have been demonstrated to cause autoimmunity in specific crosses (*Barragan et al., 2021*), suggesting that in other cases FPA overexpression could trigger or enhance autoimmune phenotypes. Consequently, modulation of transcription termination may shift the balance of costs and benefits associated with NLR gene expression. This phenomenon is not likely to be restricted to FPA because mutations in the RNA 3′ processing factor CPSF30 can also suppress autoimmunity (*Bruggeman et al., 2014*).

The impact of FPA overexpression on gene expression and immunity revealed here derives from artificial transgene expression. However, pathogens could similarly modulate NLR activity by evolving effectors that target the expression or activity of factors controlling NLR poly(A) site choice. Consistent with this idea, the HopU1 effector of *P. syringae* targets the RNA-binding protein AtGRP7 (*Fu et al., 2007*), which co-purified with FPA. In addition, the Pi4089 effector of the oomycete pathogen *Phytophthora infestans* targets the KH-domain RNA-binding protein StKRBP1 in potato; as a result, the abundance of StKRBP1 increases and infection by *P. infestans* is enhanced (*Wang et al., 2015*). This precedent reveals that effector-mediated increases in RNA-binding protein abundance can transform host RNA-binding proteins into susceptibility factors. Phylogenetic analysis of StKRBP1 suggests that a direct homolog is absent in Brassicaceae. However, the most closely related Arabidopsis proteins are FLK and PEP (*Zhang et al., 2020a*), both of which co-purify with FPA and have been shown to regulate poly(A) site choice (*Rodríguez-Cazorla et al., 2015*). FPA, GRP7, FLK and PEP, along with other RNA-binding proteins, act in concert to fine-tune the timing of flowering through the regulation of *FLC*. In a similar way, RNA-binding protein-dependent modulation of NLR expression might explain how quantitative disease resistance occurs (*Corwin and Kliebenstein, 2017*).

## New ways to analyse RNA processing

An essential feature of our study was the introduction of new approaches to study RNA processing and 3′ end formation. The use of long-read Nanopore DRS transformed our understanding of the complexity of NLR gene expression by providing insight that short-read Illumina RNA-Seq and Helicos DRS could not. We recently showed that Nanopore DRS mapping of RNA 3′ ends closely agrees with short-read Helicos DRS, and that Nanopore DRS is not compromised by internal priming artefacts (*Parker et al., 2020*). Consequently, we used Nanopore DRS to quantify alternative patterns of cleavage and polyadenylation. We also introduced a new approach to analyse alternative polyadenylation by applying the EMD metric. EMD incorporates information on the both the relative abundance and the genomic distance between alternative poly(A) sites. This is valuable because large distances between poly(A) sites are more likely to impact the mRNA coding potential or trigger mRNA surveillance compared with subtle changes in 3′UTR length.

A limitation of short-read analyses of RNA processing is their dependence upon reference transcript annotations because these may be incomplete. For example, in disease or mutant conditions, RNA processing often occurs at novel sites that are not present in reference transcriptomes (as was the case here for NLR genes). For this reason, using long-read sequencing data to generate bespoke reference transcriptomes for the genotypes under analysis can increase the value of short-read sequencing data. Until the throughput of long-read sequencing matches that of short-read technologies, a combined approach is likely to be generally useful in interpreting transcriptomes.

## Concluding remarks

It is difficult to identify alternative polyadenylation from conventional short-read RNA-Seq data. As a result, the impact of alternative polyadenylation is probably under-reported. Here we show that premature transcription termination of NLR genes is widespread. Using Nanopore DRS, we could improve the accuracy of NLR annotation and revealed a layer of NLR gene regulation that may also influence the dynamics of NLR evolution. The continued development of approaches that reveal full-length native RNA molecules is likely to provide new insight into other important, but previously unrecognised, aspects of biology.

# Materials and methods

## Key resources table

| Reagent type (species) or resource | Designation | Source or reference | Identifiers | Additional information |
|---|---|---|---|---|
| Strain (*Arabidopsis thaliana*) | Columbia (Col-0) | NA | ABRC: CS22625 | Country of Origin: USA |
| Strain (*Arabidopsis thaliana*) | Keswick (Ksk-1) | *Lai and Eulgem, 2018* | ABRC: CS1634 | Country of Origin: UK |
| Gene (*Arabidopsis thaliana*) | *FPA* | NA | TAIR/ABRC: AT2G43410 | - |
| Gene (*Arabidopsis thaliana*) | *RPP7* | NA | TAIR/ABRC: AT1G58602 | - |
| Genetic reagent (*Arabidopsis thaliana*) | *fpa-7* | *Duc et al., 2013* | ABRC: SALK_021959C | T-DNA insertion mutant in Col-0 background. Gifted by R. Amasino, UW-Madison. |
| Genetic reagent (*Arabidopsis thaliana*) | *fpa-8* | *Bäurle et al., 2007* | TAIR: 4515120225 | EMS point mutation in Col-0 background. Gifted by C. Dean, John Innes Centre |
| Genetic reagent (*Arabidopsis thaliana*) | *35S::FPA:YFP fpa-8* | *Bäurle et al., 2007* | NA | Transgenic line in *fpa-8* background, gifted by C. Dean, John Innes Centre |
| Genetic reagent (*Arabidopsis thaliana*) | *pFPA::FPA fpa-8* | *Zhang et al., 2016* | NA | Transgenic line in *fpa-8* background. |
| Genetic reagent (*Arabidopsis thaliana*) | *vir-1* | *Růžička et al., 2017* | TAIR: 6532672723 | EMS point mutant in Col-0 background. Gifted by K. Růžička, Brno. |
| Commercial assay, kit | Rneasy Plant Mini kit | QIAGEN | Cat#: 74904 | - |
| Commercial assay, kit | SuperScript III Reverse Transcriptase | Thermo Fisher Scientific | Cat#: 18080044 | - |
| Commercial assay, kit | NEBNext Ultra Directional RNA Library Prep Kit for Illumina | New England Biolabs | Cat#: E7420 | - |
| Commercial assay, kit | Dynabeads mRNA Purification Kit | Thermo Fisher Scientific | Cat#: 61006 | - |
| Commercial assay, kit | Nanopore Direct RNA sequencing kit | Oxford Nanopore Technologies | Cat#: SQK-RNA001 | - |
| Commercial assay, kit | MinION Flow cell r9.4 | Oxford Nanopore Technologies | Cat#: FLO-MIN106 | - |
| Peptide, recombinant protein | T4 DNA ligase | New England Biolabs | Cat#: M0202 | - |
| Commercial assay, kit | Quick Ligase reaction buffer | New England Biolabs | Cat#: B6058S | - |
| Commercial assay, kit | Agencourt RNAclean XP magnetic beads | Beckman Coulter | Cat#: A63987 | - |
| Commercial assay, kit | Qubit RNA BR Assay Kit | Thermo Fisher Scientific | Cat#: Q10210 | - |
| Commercial assay or kit | RNA ScreenTape System | Agilent | Cat#: 5067–5576 - 5067–5578 | - |
| Antibody | FPA antibody | Covance | NA | Rabbit polyclonal antibody. Raised against FPA amino acids 536–901. |
| Chemical compound | [γ–32P]-ATP | Perkin Elmer | Cat#: BLU012H250UC | - |
| Commercial assay or kit | DECAprime II DNA labelling kit | Thermo Fisher Scientific | Cat#: AM1455 | - |
| Commercial assay or kit | Illustra MicroSpin G-50 Columns | GE Healthcare | Cat#: 27-5330-01 | - |

*Continued on next page*

*Continued*

| Reagent type (species) or resource | Designation | Source or reference | Identifiers | Additional information |
|---|---|---|---|---|
| Commercial assay or kit | RiboRuler High Range RNA Ladder | Thermo Fisher Scientific | Cat#: SM1821 | - |
| Peptide, recombinant protein | FastAP Thermosensitive Alkaline Phosphatase | Thermo Fisher Scientific | Cat#: EF0651 | - |
| Peptide, recombinant protein | T4 Polynucleotide Kinase | Thermo Fisher Scientific | Cat#: EK0031 | - |
| Peptide, recombinant protein | Nuclease P1 | Merck | Cat#: N8630-1VL | - |
| Peptide, recombinant protein | Calf Intestinal Alkaline Phosphatase | New England Biolabs | Cat#: M0290S | - |
| Chemical compound | N6-Methyladenosine (m6A), Modified adenosine analog | Abcam | Cat#: ab145715 | - |
| Chemical compound | Adenosine, Endogenous P1 receptor agonist | Abcam | Cat#: ab120498 | - |
| Commercial assay or kit | GFP-Trap Agarose | Chromotek | Cat#: gta-20 | - |
| Software, algorithm | d3pendr | 10.5281/zenodo.4319112 | NA | Scripts to perform differential 3' end analysis using Nanopore DRS or Helicos DRS data |
| Software, algorithm | Simpson_Barton_FPA_NLRs | 10.5281/zenodo.4319108 | NA | All pipelines, scripts and notebooks used for analyses in this manuscript. |

## Plants

### Plant material and growth conditions

The wild-type Col-0 accession and *fpa-7* were obtained from the Nottingham Arabidopsis Stock Centre. The *fpa-8* mutant (Col-0 background) and *35S::FPA:YFP* in *fpa-8* (*Bäurle et al., 2007*) were provided by C. Dean (John Innes Centre). Generation of the *pFPA::FPA* line was previously described (*Zhang et al., 2016*). Surface-sterilised seeds were sown on MS10 medium plates containing 2% agar, stratified at 4°C for 2 days, germinated in a controlled environment at 20°C under 16 hr light/8 hr dark conditions and harvested 14 days after transfer to 20°C.

## IVI-MS

### Preparation of IVI-MS samples

Seedlings were harvested 14 days after germination and cross-linked with 1% (v/v) formaldehyde under vacuum. The cross-linking reaction was stopped after 15 min by the addition of glycine to a final concentration of 0.125 M and returned to vacuum for a further 5 min. Nuclei were isolated from frozen ground plant tissue using Honda buffer (20 mM Hepes-KOH pH 7.4, 10 mM MgCl$_2$, 440 mM sucrose, 1.25% (w/v) Ficoll, 2.5% (w/v) Dextran T40, 0.5% (v/v) Triton X-100, 5 mM DTT, 1 mM PMSF, 1% (v/v) Protease Inhibitor Cocktail; (Sigma)) and collected by centrifugation at 2000 g for 17 min at 4°C. Nuclei were washed twice with Honda buffer (centrifugation at 1500 g for 15 min at 4°C between washes) and lysed in nuclear lysis buffer (50 mM Tris-HCl pH 8, 10 mM EDTA, 1% (w/v) SDS, 1 mM PMSF, 1% (v/v) Protease Inhibitor Cocktail) by sonication for four cycles of 30 s pulses with low power and 60 s cooling between pulses using a Bioruptor UCD-200 (Diagenode). Following centrifugation (16,100 g for 10 min at 4°C), the supernatant was diluted 10-fold with sample dilution buffer (16.7 mM Tris-HCl pH 8, 167 mM NaCl, 1.1% (v/v) Triton X-100, 1% (v/v) Protease Inhibitor Cocktail). Cross-linked protein complexes were isolated with GFP-trap agarose beads (Chromotek) and incubated at 4°C with constant rotation for 5 hr, followed by centrifugation (141 g for 3 min at 4°C). Beads were washed three times with washing buffer (150 mM NaCl, 20 mM Tris-HCl pH 8, 2 mM EDTA pH 8, 1% (v/v) Triton X-100, 0.1% (w/v) SDS, 1 mM PMSF) by centrifugations between washes (400 g for 3 min at 4°C). Samples were incubated at 90°C for 30 min to reverse the cross-linking prior to SDS-PAGE. Each biological replicate was separated into five fractions following SDS-PAGE, subjected to in-gel digestion with trypsin and submitted for LC-MS/MS analysis (LTQ Orbitrap Velos Pro

mass spectrometer; Thermo Fisher Scientific). Three biological replicates were performed for each genotype.

### IVI-MS data analysis

Raw peptide data files from *IVI-MS* were analysed by MaxQuant software (version 1.6.10.43) (*Cox and Mann, 2008*). Peptide tables were then loaded using Proteus (version 0.2.14) (*Gierlinski et al., 2018*) and summarised to protein level counts using the hi-flyer method (mean of the top three most abundant peptides). Because wild-type plants lacking GFP were used as controls, a large number of the proteins enriched by immunoprecipitation were below the detection threshold in the control. This group of proteins can be classified as 'missing not at random' (MNAR). In all proteomics experiments, there will also be a number of proteins which are not detected purely by chance: these are referred to as 'missing at random' (MAR). We treated proteins that were missing from all replicates of a condition as MNAR, and proteins that were missing only from a subset of replicates as MAR. Using the imputeLCMD package (version 2.0) (*Lazar, 2015*), a K nearest neighbours' strategy was used to impute MAR examples, and a quantile regression imputation of left centred data (QRILC) approach was used to impute MNAR examples. Differential expression analysis was performed on imputed data using limma (version 3.40.0) (*Ritchie et al., 2015*). Because imputation is not deterministic (i.e. will lead to different outcomes every time), we improved the robustness of the analysis by performing 999 bootstraps of the imputation and differential expression, and summarising the results using the median $\log_2$ fold change and harmonic mean p value.

## ChIP-Seq

### Preparation of libraries for ChIP-Seq

ChIP against FPA and Pol II phosphorylated at either Ser[5] or Ser[2] of the CTD heptad repeat was performed as previously described (*Yu et al., 2019*). Polyclonal antibodies against FPA amino acids 536–901 were raised in rabbit by Covance.

### ChIP-Seq data processing

FPA and Pol II ChIP-Seq data are available at ENA accession PRJNA449914. H3K9me[2] ChIP-Seq data were downloaded from ENA accessions PRJDB5192 (*Inagaki et al., 2017*) and PRJNA427432 (*Lai et al., 2020*). Reads were aligned to the TAIR10 reference genome using Bowtie2 (version 2.3.5.1) (*Langmead and Salzberg, 2012*) with the parameters –mm –very-sensitive –maxins 800 –no-mixed –no-discordant. Counts per million normalised coverage profiles were generated using deepTools (version 3.4.3) (*Ramírez et al., 2014*). For 3′ end centred metagene profiles, we determined the major 3′ position per gene using the Araport11 annotation and existing Col-0 Helicos DRS data (*Sherstnev et al., 2012*). Metagenes centred on these positions were then generated in Python 3.6 using pyBigWig (version 0.3.17) (*Ramírez et al., 2014*), Numpy (version 1.18.1) (*Harris et al., 2020*) and Matplotlib (version 3.1.3) (*Hunter, 2007*). For differential H3K9me[2] analysis, read counts per gene (including intronic regions) were generated using pysam (version 0.16.0), and differential expression analysis was performed using edgeR (version 3.22.5) (*Robinson et al., 2010*).

## RNA

### Total RNA isolation

Total RNA was isolated using RNeasy Plant Mini kit (QIAGEN) and treated with TURBO DNase (Thermo Fisher Scientific) according to the manufacturers' instructions. The total RNA concentration was measured using a Qubit 1.0 Fluorometer and Qubit RNA BR Assay Kit (Thermo Fisher Scientific), whilst RNA quality and integrity was assessed using a NanoDrop 2000 spectrophotometer (Thermo Fisher Scientific) and Agilent 2200 TapeStation System (Agilent).

## Nanopore DRS

### Preparation of libraries for DRS using nanopores

Total RNA was isolated from Col-0, *fpa-8* and *35S::FPA:YFP* seedlings as described above. mRNA was isolated and Nanopore DRS libraries prepared (using the SQK-RNA001 Nanopore DRS Kit; Oxford Nanopore Technologies) as previously described (*Parker et al., 2020*). Libraries were loaded

onto R9.4 flow cells (Oxford Nanopore Technologies) and sequenced using a 48 hr runtime. Four biological replicates were performed for each genotype.

## Nanopore DRS data processing

Nanopore DRS reads were basecalled using the Guppy (version 3.6.0) high-accuracy model. Reads were mapped to the Arabidopsis *TAIR10* genome (*Arabidopsis Genome Initiative, 2000*) using minimap2 (version 2.17) with the parameters -a -L –cs=short x splice -G20000 –end-seed-pen=12 –junc-bonus=12 uf. Spliced alignment was guided using junctions from the Araport11 annotation (*Cheng et al., 2017*). Nanopore DRS reads can suffer from 'oversplitting' – where the signal originating from a single RNA molecule is incorrectly interpreted as two or more reads (*Parker et al., 2020*). These errors can be systematic and result in false positive 3′ ends. To filter these errors, we identified reads that were sequenced consecutively through the same pore and also mapped contiguously on the genome (within 1 kb of each other). In this way, we filtered all except the most 3′ reads, which should contain the genuine RNA 3′ end. Pipelines for processing Nanopore DRS data were built using Snakemake (*Köster and Rahmann, 2012*).

## Helicos DRS
## Preparation of samples for Helicos DRS

Total RNA was isolated from the Col-0, *fpa-8* and *35S::FPA:YFP* seedlings as described above. Samples were processed by Helicos BioSciences as previously described (*Ozsolak et al., 2009*; *Sherstnev et al., 2012*). Three biological replicates were performed for each genotype.

## Helicos DRS data processing

Helicos DRS reads were mapped to the *Arabidopsis* TAIR10 genome using Heliosphere (version 1.1.498.63) as previously described (*Sherstnev et al., 2012*). Reads were filtered to remove those with insertions or deletions of >4 nt and to mask regions with low complexity, as determined using DustMasker (*Camacho et al., 2009*) (from BLAST +suite version 2.10.1) set at DUST level 15 (*Sherstnev et al., 2012*).

## Differential 3′ end analysis of Nanopore and Helicos DRS datasets

Transcriptional loci were first identified in Col-0, *fpa*-8 and *35S::FPA:YFP* Nanopore DRS reads using the long-read transcript assembly tool StringTie2 version 2.1.1 (*Pertea et al., 2015*). Novel transcriptional loci were merged with annotated loci from the Araport11 reference (*Cheng et al., 2017*). To detect sites with altered 3′ end distributions in *fpa-8* and *35S::FPA:YFP*, we pooled the replicates of either Nanopore or Helicos DRS data and identified reads overlapping each transcriptional locus. These reads were used to build distributions of 3′ end locations. The difference in 3′ end distributions between the treatment and control (Col-0) was measured using EMD. To identify loci with statistically significant differences in 3′ distributions, we performed an EMD permutation test using 999 bootstraps: for this, reads for each locus were randomly shuffled between the treatment and control samples to create null distributions, and the EMD recalculated. The histogram of null EMDs was fitted using a gamma distribution, and the p-value (probability of achieving the observed EMD or greater by chance) was calculated from the distribution. p-Values were corrected for multiple testing using the Benjamini–Hochberg method. Genes with an EMD >25 and an FDR < 0.05 were considered to be differentially alternatively polyadenylated, and the directionality of change was identified using the difference in mean 3′ position. Software developed to perform differential 3′ analysis is available on GitHub at https://github.com/bartongroup/d3pendr and 10.5281/zenodo.4319113, and can be used with Nanopore DRS, Helicos DRS, or Illumina 3′ tag-based datasets.

## Illumina RNA sequencing
## Preparation of libraries for Illumina RNA sequencing

Total RNA was isolated from the Col-0, *fpa-8* and *35S::FPA:YFP* seedlings as described above. mRNA was isolated and sequencing libraries prepared using the NEBNext Ultra Directional RNA Library Prep Kit for Illumina (New England Biolabs) by the Centre for Genomic Research (University of Liverpool). 150 bp paired-end sequencing was carried out on Illumina HiSeq 4000. Six biological replicates were performed for each genotype.

### Illumina RNA sequencing data processing

Illumina RNA-Seq data were assessed for quality using FastQC (version 0.11.9) and MultiQC (version 1.8) (*Andrews, 2010*; *Ewels et al., 2016*). Reads were mapped to the TAIR10 genome using STAR (version 2.7.3a) (*Dobin et al., 2013*) with a splice junction database generated from the Araport11 reference annotation (*Cheng et al., 2017*). Counts per million normalised coverage tracks were created using samtools (version 1.10) and deepTools (version 3.4.3) (*Ramírez et al., 2014*). To identify expressed regions in each locus, the coverage profiles of each treatment and control replicate were first extracted using pyBigWig (version 0.3.17) (*Ramírez et al., 2014*). These were normalised such that the area under each profile was equal to the mean area under the profiles. A normalised coverage threshold of 1 was used to identify expressed regions of the loci. These regions were further segmented when at least two-fold differences in expression within a 25-nt window were found between control and treatment conditions (and then regions smaller than 50 nt removed). Expression of the segmented regions was then calculated using featureCounts (version 2.0.0) (*Liao et al., 2013*). Each read pair was counted as one fragment, and only properly paired, concordant and primary read pairs were considered. Differential usage within transcriptional loci was assessed using DEXSeq (version 1.32.0) (*Reyes et al., 2013*). Loci were considered to be differentially processed if they had a locus-level FDR < 0.05 and at least one region with an absolute logFC >1 and FDR < 0.05. For differential splice junction usage analysis, counts of splice junctions annotated in the bespoke Nanopore DRS-derived annotation, plus Araport11 annotation, were generated for each locus using pysam (version 0.16.0). Differential splice junction usage was assessed using DEXSeq (version 1.32.0) (*Reyes et al., 2013*). Loci were considered to be differentially spliced if they had a locus-level FDR < 0.05 and at least one junction with an absolute logFC >1 and FDR < 0.05.

### Gene tracks

Gene track figures were generated in Python 3.6 using Matplotlib (version 3.1.3) (*Hunter, 2007*). For gene tracks where any condition had >200 Nanopore DRS read alignments, 200 representative alignments were selected by random sampling without replacement (except for the *FPA* gene track figure, where 500 read alignments were sampled). nanoPARE data (*Schon et al., 2018*) were processed as previously described (*Parker et al., 2020*). For reannotated gene loci, domains were predicted using the InterproScan web client (*Mitchell et al., 2019*) and LRRs were predicted using LRRpredictor web client (*Martin et al., 2020*). Protein alignments were created and visualised in Jalview (version 2.11) (*Waterhouse et al., 2009*) using T-Coffee (*Notredame et al., 2000*).

### Protein domain family enrichment analysis

To conduct protein domain enrichment analysis, InterPro domain annotations of Arabidopsis proteins were downloaded from BioMart (*Smedley et al., 2009*) and converted to genomic co-ordinates using the Araport11 annotation (*Cheng et al., 2017*). Domain families overlapping each locus tested for alternative polyadenylation using either Nanopore or Helicos DRS were identified using pybedtools (version 0.8.1) (*Dale et al., 2011*). To identify enriched domain families, domains were randomly shuffled between tested loci in 19,999 bootstraps, and the number of times that each domain class overlapped by chance with significantly alternatively polyadenylated loci was recorded. This was compared with the observed overlap of each domain family with alternatively polyadenylated loci to calculate p-values, which were corrected for multiple testing using the Benjamini–Hochberg method.

### Manual annotation of alternatively polyadenylated NLR genes

To identify which of the 206 previously annotated NLR genes present in Col-0 were alternatively polyadenylated in *fpa-8* and *35S::FPA*, we devised a standard operating procedure for visual inspection. Genes that had Nanopore DRS read coverage in at least two conditions were considered to be expressed. Genes were considered to be alternatively polyadenylated if they had multiple 3′ end locations with each supported by at least four Nanopore DRS reads, and if there was a clear difference in Nanopore DRS coverage in the treatment condition compared with Col-0. Helicos and Illumina corroboration of poly(A) sites and coverage changes was also taken into consideration.

## Genomic organisation of alternatively polyadenylated NLR genes

To test whether expressed NLR genes with FPA-dependent alternative polyadenylation were associated with NLR gene clusters, we used previously produced cluster assignments for Col-0 NLR genes (*Lee and Chae, 2020*). We also tested the association of FPA-dependent alternative polyadenylation with previously produced hypervariable NLR classifications (*Prigozhin and Krasileva, 2021*). The association of alternatively polyadenylated genes with both major NLR gene clusters and hypervariable NLRs was assessed using a Chi squared test. To test whether FPA-sensitive NLRs are found in regions with high synteny diversity, we used 5 kb sliding window estimates of synteny diversity calculated from seven diverse Arabidopsis ecotypes (*Jiao and Schneeberger, 2020*). For each expressed NLR gene, the window with the largest overlap was used as the estimate of synteny diversity. The association with alternatively polyadenylated genes was assessed using a *t*-test.

## RNA gel blot analysis of *RPP7* mRNAs

RNA gel blot analysis was carried out as previously described (*Quesada et al., 2003*) with minor modifications. *RPP7* mRNA was detected using a probe annealing to the second exon of the *RPP7* (*AT1G58602*) gene (200 bp PCR product amplified with the following primers: Forward: 5′-TCGGGGACTACTACTACTCAAGA-3′ and Reverse: 5′-TCTTGATGGTGTGAAAGAATCTAGT-3′). *β-TUBULIN* mRNA was used as a loading control and visualised by a probe annealing to the third exon of the *β-TUBULIN (AT1G20010) gene* (550 bp PCR product amplified with the following primers: Forward: 5′- CTGACCTCAGGAAACTCGCG-3′ and Reverse: 5′- CATCAGCAGTAGCATCTTGG-3′). The probes were 5′ labelled using [γ-$^{32}$P]-ATP (Perkin Elmer) and DECAprime II DNA labelling kit (Thermo Fisher Scientific) and purified on illustra G-50 columns (GE Healthcare Life Sciences). mRNA isoforms were visualised and quantified using an Amersham Typhoon Gel and Blot Imaging System (GE Healthcare Bio-Sciences AB). The RiboRuler High Range RNA Ladder (Thermo Fisher Scientific), used to identify the approximate size of RNA bands, was first dephosphorylated using FastAP Thermosensitive Alkaline Phosphatase (Thermo Fisher Scientific) and then labelled with [γ-$^{32}$P]-ATP (Perkin Elmer) using T4 Polynucleotide Kinase (Thermo Fisher Scientific) before gel loading.

## m$^6$A LC-MS/MS

Total RNA was isolated and checked as described above. mRNA was extracted twice from approximately 75 µg of total RNA using the Dynabeads mRNA Purification Kit (Thermo Fisher Scientific) according to the manufacturer's instructions. The quality and quantity of mRNA was assessed using a NanoDrop 2000 spectrophotometer (Thermo Fisher Scientific) and Agilent 2200 TapeStation System (Agilent). Samples for m$^6$A LC-MS/MS were prepared as previously described (*Huang et al., 2018*) with several modifications. First, 100 ng mRNA was diluted in a total volume of 14 ml nuclease-free water (Thermo Fisher Scientific) and digested by nuclease P1 (1 U, Merck) in 25 µl buffer containing 20 mM $NH_4OAc$ (pH 5.3) at 42°C for 2 hr. Next, 3 µl freshly made 1 M $NH_4HCO_3$ and calf intestinal alkaline phosphatase (1 U, New England Biolabs) were added, and samples were incubated at 37°C for 2 hr. The samples were then diluted to 50 µl with nuclease-free water and filtered (0.22 µm pore size, 4 mm diameter; Millipore). LC-MS/MS was carried out by the FingerPrints Proteomics facility at the University of Dundee. m$^6$A/A ratio quantification was performed in comparison with the curves obtained from pure adenosine (endogenous P1 receptor agonist, Abcam) and m$^6$A (modified adenosine analog, Abcam) nucleoside standards. Statistical analysis was performed using a two-way *t*-test.

## Pathogenesis assays

Pathogenesis assays were carried out as previously described (*Tomé et al., 2014*). The *Hpa*-Hiks1 isolate was maintained by weekly sub-culturing on Ksk-1 plants. A solution containing *Hpa*-Hiks1 spores was used to inoculate 14-day-old Col-0, Ksk-1, *fpa-7*, *fpa-8*, *pFPA::FPA*, and *35S::FPA:YFP* seedlings. Sporangiophores were counted 4 days after inoculation. The experiment was repeated three times using up to 45 plants per genotype per each repeat. Statistical analysis was performed with negative binomial regression using Statsmodels (version 0.11.0) (*Seabold and Perktold, 2010*), plants were grouped by experimental repeat during testing to control for variation between repeats.

## Code availability

All pipelines, scripts and notebooks used to generate figures are available from GitHub at https://github.com/bartongroup/Simpson_Barton_FPA_NLRs and Zenodo at 10.5281/zenodo.4319108. The software tool developed for detecting changes in poly(A) site choice in Nanopore and Helicos DRS data are available from GitHub at https://github.com/bartongroup/d3pendr and Zenodo at 10.5281/zenodo.4319112.

## Acknowledgements

We thank Paul Birch and Ingo Hein for comments on the manuscript and David Baulcombe, Ian Henderson and Wenbo Ma for helpful NLR discussions. We thank Abdelmadjid Atrih (Centre for Advanced Scientific Technologies, School of Life Sciences) for the $m^6A$ LC-MS/MS analysis. This work was supported by awards from the BBSRC (BB/M010066/1; BB/J00247X/1; BB/M004155/1), the University of Dundee Global Challenges Research Fund, a University of Dundee PhD studentship to V.Z. and a European Union Horizon 2020 research and innovation programme under Marie Skłodowska-Curie grant agreement No. 799300 to KK. P.G.P.M. received the support of the EU in the framework of the Marie-Curie FP7 COFUND People Programme, through the award of an AgreenSkills+ fellowship ( grant agreement no. 609398). This work was supported by a grants to S. D.M from the National Institutes of Health (GM075060) and National Science Foundation (2001115). The FingerPrints Proteomics Facility of the University of Dundee is supported by a Wellcome Trust Technology Platform Award (097945/B/11/Z).

## Additional information

### Funding

| Funder | Grant reference number | Author |
| --- | --- | --- |
| Biotechnology and Biological Sciences Research Council | BB/M010066/1 | Geoffrey J Barton Gordon G Simpson |
| Biotechnology and Biological Sciences Research Council | BB/J00247X/1 | Geoffrey J Barton Gordon G Simpson |
| Biotechnology and Biological Sciences Research Council | BB/M004155/1 | Geoffrey J Barton Gordon G Simpson |
| H2020 Marie Skłodowska-Curie Actions | 799300 | Katarzyna Knop |
| Wellcome Trust | 097945/B/11/Z | Geoffrey John Barton Gordon G Simpson |
| National Institutes of Health | GM075060 | Scott D Michaels |
| FP7-PEOPLE | 609398 | Pascal GP Martin |
| National Science Foundation | 2001115 | Scott D Michaels |

The funders had no role in study design, data collection and interpretation, or the decision to submit the work for publication.

### Author contributions

Matthew T Parker, Conceptualization, Resources, Data curation, Software, Formal analysis, Supervision, Funding acquisition, Validation, Investigation, Visualization, Methodology, Writing - original draft, Project administration, Writing - review and editing; Katarzyna Knop, Conceptualization, Data curation, Software, Formal analysis, Funding acquisition, Validation, Investigation, Visualization, Methodology, Writing - original draft, Project administration, Writing - review and editing; Vasiliki Zacharaki, Conceptualization, Funding acquisition, Validation, Investigation, Visualization, Methodology, Writing - original draft, Project administration, Writing - review and editing; Anna V Sherwood, Conceptualization, Validation, Investigation, Methodology, Project administration, Writing - review and editing; Daniel Tomé, Conceptualization, Formal analysis, Validation, Investigation, Methodology, Project administration, Writing - review and editing; Xuhong Yu, Formal analysis, Validation,

Investigation, Methodology, Writing - review and editing; Pascal GP Martin, Data curation, Formal analysis, Validation, Investigation, Methodology; Jim Beynon, Resources, Formal analysis, Supervision, Investigation, Project administration; Scott D Michaels, Conceptualization, Resources, Supervision, Project administration; Geoffrey J Barton, Conceptualization, Resources, Supervision, Funding acquisition, Project administration, Writing - review and editing; Gordon G Simpson, Conceptualization, Resources, Supervision, Funding acquisition, Writing - original draft, Project administration, Writing - review and editing

### Author ORCIDs

Matthew T Parker (iD) https://orcid.org/0000-0002-0891-8495
Katarzyna Knop (iD) http://orcid.org/0000-0002-2636-9450
Vasiliki Zacharaki (iD) http://orcid.org/0000-0002-5543-2332
Pascal GP Martin (iD) https://orcid.org/0000-0002-4271-658X
Scott D Michaels (iD) https://orcid.org/0000-0001-5248-3487
Geoffrey J Barton (iD) https://orcid.org/0000-0002-9014-5355
Gordon G Simpson (iD) https://orcid.org/0000-0001-6744-5889

### Decision letter and Author response

Decision letter https://doi.org/10.7554/eLife.65537.sa1
Author response https://doi.org/10.7554/eLife.65537.sa2

## Additional files

### Supplementary files

• Supplementary file 1. Proteins co-purifying with FPA, as identified by *IVI-MS* [Linked to *Figure 1*].

• Supplementary file 2. Properties of the sequencing datasets produced using Nanopore DRS, Helicos DRS and Illumina RNA-Seq [Linked to *Figure 2*].

• Transparent reporting form

### Data availability

IVI-MS data is available from the ProteomeXchange Consortium via the PRIDE partner repository with the dataset identifier PXD022684 (Perez-Riverol et al., 2019, DOI: https://doi.org/10.1093/nar/gky1106). FPA and Pol II ChIP-Seq data is available from ENA accession PRJNA449914. Col-0 nanopore DRS data is available from ENA accession PRJEB32782. fpa-8 and 35S::FPA:YFP nanopore DRS data is available from ENA accession PRJEB41451. hen2-2 nanopore DRS data is available from ENA accession PRJEB41381. Col-0, fpa-8 and 35S::FPA:YFP Helicos DRS data is available from Zenodo DOI 10.5281/zenodo.4309752. Col-0, fpa-8 and 35S::FPA:YFP Illumina RNA-Seq data is available from ENA accession PRJEB41455. All pipelines, scripts and notebooks used to generate figures are available from GitHub at https://github.com/bartongroup/Simpson_Barton_FPA_NLRs and Zenodo a thttps://doi.org/10.5281/zenodo.4319109. The software tool developed for detecting changes in poly(A) site choice in Nanopore and Helicos DRSdata are available from GitHub at https://github.com/bartongroup/d3pendr and Zenodo athttps://doi.org/10.5281/zenodo.4319113.

The following datasets were generated:

| Author(s) | Year | Dataset title | Dataset URL | Database and Identifier |
|---|---|---|---|---|
| Parker MT, Knop K, Zacharaki V, Sherwood AV, Tome D, Yu X, Martin P, Beynon J, Michaels S, Barton GJ, Simpson GG | 2020 | Proteomic analysis of FPA interactors | https://www.ebi.ac.uk/pride/archive/projects/PXD022684 | PRIDE, PXD022684 |
| Parker MT, Knop K, Zacharaki V, Sherwood AV, | 2020 | Nanopore direct RNA sequencing of FPA mutants and overexpressors | https://www.ebi.ac.uk/ena/browser/view/PRJEB41451 | ENA, PRJEB41451 |

| Tome D, Yu X, Martin P, Beynon J, Michaels S, Barton GJ, Simpson GG | | | | | |
|---|---|---|---|---|---|
| Parker MT, Knop K, Zacharaki V, Sherwood AV, Tome D, Yu X, Martin P, Beynon J, Michaels S, Barton GJ, Simpson GG | 2020 | Helicos direct RNA sequencing of FPA mutants and overexpressors | https://zenodo.org/record/4309752 | Zenodo, 10.5281/zenodo/4309752 | |
| Parker MT, Knop K, Zacharaki V, Sherwood AV, Tome D, Yu X, Martin P, Beynon J, Michaels S, Barton GJ, Simpson GG | 2020 | Illumina RNA-Seq of FPA mutants and overexpressors | https://www.ebi.ac.uk/ena/browser/view/PRJEB41455 | ENA, PRJEB41455 | |
| Matthew T Parker | 2020 | bartongroup/Simpson_Barton_FPA_NLRs: preprint version | https://doi.org/10.5281/zenodo.4319109 | Zenodo, 10.5281/zenodo.4319109 | |
| Matthew T Parker | 2020 | bartongroup/d3pendr: preprint release | https://doi.org/10.5281/zenodo.4319113 | Zenodo, 10.5281/zenodo.4319113 | |

The following previously published datasets were used:

| Author(s) | Year | Dataset title | Dataset URL | Database and Identifier |
|---|---|---|---|---|
| Parker MT, Knop K, Barton GJ, Simpson GG | 2020 | Nanopore direct RNA sequencing of hen2-2 mutants | https://www.ebi.ac.uk/ena/browser/view/PRJEB41381 | ENA, PRJEB41381 |
| Parker MT, Knop K, Zacharaki V, Sherwood AV, Tome D, Yu X, Martin P, Beynon J, Michaels S, Barton GJ, Simpson GG | 2020 | Nanopore Direct RNA Sequencing Maps the Arabidopsis m6A Epitranscriptome | https://www.ebi.ac.uk/ena/browser/view/PRJEB32782 | ENA, PRJEB32782 |
| Yu X, Martin PGP, Michaels SD | 2019 | Genome-wide occupancy of BDR1, BDR2 and FPA (ChIP-seq) | https://www.ebi.ac.uk/ena/browser/view/PRJNA449914 | ENA, PRJNA449914 |
| Schon MA, Kellner MJ, Plotnikova A, Hofmann F, Nodine MD | 2018 | nanoPARE: Parallel analysis of RNA 5' ends from low input RNA | https://www.ebi.ac.uk/ena/browser/view/PRJNA449355 | ENA, PRJNA449355 |
| Inagaki S, Takahashi M, Hosaka A, Ito T, Toyoda A, Fujiyama A, Tarutani Y, Kakutani T | 2017 | The gene-body chromatin modifications dynamics mediates epigenome differentiation in Arabidopsis | https://www.ebi.ac.uk/ena/browser/view/PRJDB5192 | ENA, PRJDB5192 |
| Lai Y, Lu XM, Daron J, Pan S, Wang J, Wang W, Tsuchiya T, Holub E, McDowell JM, Slotkin RK, Le RKG, Eulgem T | 2020 | Genome-wide profilings of EDM2-mediated effects on H3K9me2 and transcripts in *Arabidopsis thaliana* | https://www.ebi.ac.uk/ena/browser/view/PRJNA427432 | ENA, PRJNA427432 |

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
