## [Decision Letter]

**Acceptance summary:**

In this study, the authors examined the function of the RNA-binding protein FPA through analyzing its protein interactome and its global impact on gene expression using a combined approaches of Nanopore DRS, Helicos DRS, and short-read Illumina RNA-Seq. The combined datasets and new computational approaches developed by the authors showed a predominant role of FPA in promoting poly(A) site choice. The authors further revealed that FPA mediates widespread premature cleavage and polyadenylation of the transcripts of NLR genes, which act as important plant immune regulators. Overall, this study suggests that control of transcription termination processes mediated by FPA provides an additional layer of the regulatory dynamics of NLRs in plant immune responses.

**Decision letter after peer review:**

Thank you for submitting your article “Widespread premature transcriptiontermination of *Arabidopsis thaliana* NLR genes by the spen protein FPA” for consideration by eLife. Your article has been reviewed by 3 peer reviewers, and the evaluation has been overseen by Hao Yu as the Reviewing Editor and Detlef Weigel as the Senior Editor. The following individuals involved in review of your submission have agreed to reveal their identity: Chae Eunyoung (Reviewer #1); Blake C Meyers (Reviewer #3).

Essential Revisions:

While there was agreement that the topic is timely and the findings relevant, there were some concerns regarding manuscript structure, inconsistency among some results, and interpretation of biological relevance of the data, as listed below, that need to be addressed to support the conclusions.

1. The manuscript presents an extensive body of studies in analyzing FPA interacting proteins and its potential RNA targets including NLRs. Although the overall results cover a series of observations, many of them are descriptive and divert the audience's attention from understanding the novelty and significance of the findings. Thus, we suggest that the authors re-organize the manuscript into a more coherent story and focus on the most important data pertaining to NLR control as shown in the title and the abstract.

2. The authors should address or explain some inconsistencies in the results as mentioned by reviewers. For example, the authors found that at least for some pathogens, "loss of FPA function does not reduce plant resistance". This is not consistent with the hypothesis that FPA is important for regulating NLR immune response genes, and the observation that premature exonic termination of RPP7 caused by FPA has a functional consequence for Arabidopsis immunity against Hpa-Hiks1.

3. The significance of this study will be strengthened by analysis of the biological relevancy of the alternative polyadenylation events mediated by FPA pertaining to NLR functions. We suggest that the authors consider either providing new experimental data or clearly interpreting existing results, such as those relevant to regulation of RPP7, to provide better insights into biological significance of the data presented in this manuscript.

Please also take into consideration the other specific comments from the reviewers below to revise the manuscript.

**Reviewer #1:**

The manuscript by Parker and colleagues presents an extensive body of work on characterizing the role of FPA in the choice of polyadenylation sites in transcripts of *A. thaliana*. Investigation on the mechanistic details that FPA engages on the mRNA processing was first initiated with the in vivo pull-down followed by LC-MS/MS, which revealed the its protein interactome relevant for 3'-end processing. The main dataset pertaining to the manuscript title comes from the comparative transcriptome analysis of Col-0, fpa-8 mutant and the overexpressor of FPA, 35S:FPA:YFP. The strength of this work lies in the use of nanopore DRS by demonstrating the layers of FPA-dependent transcripts, including its own, and its comparison to datasets by Illumina RNA-Seq and Helicos DRS. The systematic analysis uncovered unexpected complexity in the A. thaliana NLR transcriptome under the control of FPA and thus delivers a new insight on NLR biology. Several studies anecdotally have reported the importance of using genomic DNA, but not a single cDNA species, for addressing full functionality of NLR genes. Recent advances in NLRome sequencing from multiple genomes of a species and NLR structure/function studies also highlight the importance of understanding modular nature of NLR. As alluded with the modular diversity of NLRs kept in the genomes of a species in recent studies, NLR genes are prone to reshuffle in the genome to generate different variants, including partial entities with the loss of some parts of the proteins or even chimeras, supposedly maximizing the repertoire for defense. This work adds the level of transcript diversity on that of genomic diversity; FPA, an essential factor for transcription termination determinant, targets numerous NLRs to control the layers of NLR transcriptome of an individual plant. Although it is yet to be clarified for the regulatory significance of FPA-mediated NLR transcript changes under biotic or abiotic conditions, the authors succeeded in employing fine genetic schemes utilizing FPA-defective vs. -overexpressing lines along with long-read nanopore DRS technology for the first time to uncover the breadth of differential transcript generation focused on 3'-end choices. This work is timely and impactful for NLR research owing to the above-mentioned recent advances in NLR field.

As this work is the first of its kind in utilizing nanopore DRS to address NLR transcriptome, several technical concerns can be addressed to corroborate the claims made in the manuscript, which authors can find in the following section (1-8). Regarding the organization of the manuscript, the authors may consider to rebalance the two parts: FPA interactome vs. FPA targets and NLRs. Overall, the manuscript can be seen as combining two stories; first to characterize FPA function in 3'-end processing of transcripts inferred by interacting proteomes and meta-analysis of ChIP-seq data; second part includes detailed analysis of NLR transcripts and others. Although the first half of the analysis is a necessary prelude to the following NLR analysis, the current title and academic novelty mainly lies, or were intended by the authors, on the NLR analysis. However, current manuscript has relatively enlarged section of the first with NLR analysis packed into a series of supplementary dataset. If authors wishes to opt for highlighting NLR analysis, the following suggestions would help (9-14).

1) Earth mover distance (EMD) has been applied to identify a locus with alternative polyadenylation. What is the basis of using EMD value of 25 as a cutoff? According to Figure 4 B,D, EMD can range from 0-4000. One would also wonder if the distance unit equals bp.

In addition, EMD values of some genes (e.g. FPA and representative NLRs) can be specified in the main dataset so that significance of the cut-off values shall be appreciated.

2) Regarding the manual annotation of alternatively polyadenylated NLR genes (L1160-):

Genes with alternative polyadenylation were identified and the ending location was supported when there were minimum four DRS reads. It would be relevant to provide the significance of "the four" based on read coverage statistics, for example, with average read number covering an annotated NLR transcript with the specification of an average size.

3) Figure 4E shows that Ilumina-RNAseq dataset detects the number of loci with a different order of magnitude compared with the other two methods. Reference-agonistic pipeline shall be appreciated, however, the method engaged might have elevated the counting of paralogous reads mapped to different locations than they should be. Along with paralogous read collapsing, this is always a problem with tandemly repeated genes, such as NLRs by and large. For example, NLR paralogs in a complex cluster with conserved TIR/NBS but diversified LRRs would have higher coverage in the first two domains but drop in the diversified parts. The authors need to specify their bioinformatic consideration to avoid such problems.

Although the tone of the Illumina read section was careful and the main 3'-end processing conclusion was made by nanopore DRS, the authors are also advised to clearly state the limitation of using Illumina-RNAseq to address alternative polyadenylating sites at the beginning of the section, for example what to be maximally taken out from Figure 4 E and 4F. This will give relative weights to each dataset generated by different methods. One advantage of using Illumina data would be that the expression level changes can be associated with changes in processing, it seems.

4) At the RPP7 locus, At1g58848 is identical in sequences with At1g59218 as is At1g58807 with At1g59214 (two twins in the RPP7 cluster by tandem duplication). It would be good to check whether the TE At1g58889 readthrough indeed occurs in the sister duplicate with a potential TE in the downstream of At1g59218. If not, it can be used as an example of duplication and neofunctionalization through an alternative polyadenylation site choices.

5) HMM search shall be revisited to confirm if they are to detect the TIR domain. Given that a large proportion of NLRs in *A. thaliana* carry TIR at their N-terminal ends and the specified examples included TIR-NLR, it is surprising to see no TIR domain in Figure 5.

6) L659-668: how does the new data relate to the previously TAIR annotated At1g58602.1 vs At1g58602.2 (Figure 6, Inset 1)? It would be good to see these clearly stated in the main text as compared to newly identified ones. From the nanopore profiling, At1g58602.2 appears to be the dominant form.

7) One thing to note is that in the overexpressor of which Hiks1 R is suppressed, there was hardly any At1g58602.1 produced in addition to the large reduction of At1g58602.2. Thus, relative functional importance of the two transcripts shall be discussed in line with the Hpa resistance data. Accordingly, L740-741 phrasing shall be revised to include the possibility of absolute or relative "depletion" of functional transcript(s) contributing to the compromise in Hpa resistance.

8) It would be necessary to state in the main text the implication of phosphorylation on the two Ser residues on Pol II at L245. A clear description distinguishing the effect of the two phosphorylation and the specificity of the antibodies is desirable, as the data was interpreted as if the two sites made differences, such that Ser2 was heavily emphasized (e.g. subtitle). Albeit low level, Ser5 data also shows an overlap with FPA ChIP-seq coverage at the 3' end. If there is a statistical significance to be taken account to interpret the coverage, please state it. Given that elongation occurs progressively, I wonder how much should be taken out from the distinction.

9) Figures presentation for RPP4 and RPP7 are great in detailing the FPA-dependent NLR transcript complexity. To make the functional link more evident, the authors may consider bringing up parts of the Figure 5-supplement to a main Figure to detail the revised annotation of NLRs. Given recent advances in NLR structure and function studies, extra domain fusion, fission and truncated versions of NLRs require a great deal of attention. For example, potential functional link to the NMD-mediated autoimmunity and revised annotation of At5g46470 (RPS6) needs a clear visual guidance preferably with a main figure (Figure 5-Sup3).

10) The section "FPA controls the processing of NLR transcripts" includes dense information and can be broken down to several categories. To this end, Supplement File 3 (NLR list) shall be revised to deliver the categorical classes and further details and converted to a main table.

For NLR audience, for example, it would be important to associate the information to raw reads to assess where the premature termination would occur. At least, the ways to retrieve dataset or to curate the termination sites shall be guided.

On the contrary, there is no need to include other genes in Figure 4-Figure Supplement 4-8 under this section. They are not NLRs.

11) Figure 7 and IBM1 section can be spared to supplement.

12) The list of "truncated NLR transcripts" in particular, either by premature termination within protein-coding or with intronic polyadenylation, should be made as a main table. The table can be preferably carrying details in which degree the truncation is predicted to be made. With current sup excel files, it is difficult to assess the breadth of the FPA effect on the repertoire of NLRs and their function. This way, functional implication of differential NLRs transcriptome can be better emphasized.

13) FPA-mediated NLR transcript controls, as to promote transcript diversity, is expected to exert its maximum effect if FPA level or activity is subject to the environmental stresses, such as biotic or abiotic stresses. The discussion on effectors targeting RNA-binding proteins (L909-918) is a great attempt in broadening the impact of this research. In addition, if anything is known to modulate FPA activity, such as biotic or abiotic stresses or environmental conditions, please include in the discussion.

14) NLR transcript diversity as source of cryptic variation contributing to NLR "evolution" is an interesting concept, however, evolutionary changes require processes of genic changes affecting transcript layers or stabilizing transcriptome diversity. In the authors' proposition in looking into accessions, potential evolutionary processes can be further clarified.

**Reviewer #2:**

Parker et al attempted to show that the FPA protein functions to regulate the widespread premature transcription termination of the Arabidopsis NLR genes. Using in vivo interaction proteomic-mass spectrometry, FPA was shown to co-purified with the mRNA 3' end processing machinery. Metagene analysis was used to show that FPA co-localized with Pol II phosphorylated at Ser2 of the CTD heptad repeat at the 3' end of Arabidopsis genes. Using a combination of Illumina RNA-Seq, Helicos, and nanopore DRS technologies, FPA was found to affect RNA processing by promoting poly(A) site choice, and hence controls the processing of NLR transcripts whereas such process is independent of IBM1.

Overall, it is a potentially important research. The data is rich and could be useful. However, the biological stories described are not thoroughly supported by the data presented, especially when the authors tried to touch on several aspects without some important validations and strong connections among different parts. Some special comments are provided below:

(1) The title of this manuscript is "The expression of Arabidopsis NLR immune response genes is modulated by premature transcription termination and this has implications for understanding NLR evolutionary dynamics". Therefore, the readers will expect some functional connections between the FPA and the novel NLR isoforms due to premature transcription termination. However, the transcript levels of plant NLR genes are under strict regulation (e.g. Mol. Plant Pathol. 19:1267). Since the functions of NLR genes are related to effector-triggered immunity, it is more important to study the function of FPA on premature transcription termination when the plants are challenged with pathogens. In this manuscript, most transcript analyses are based on samples under normal growth conditions. It is therefore a weak link between the genomic studies and the functional aspects. For instance, it is more important to identify unique NLR isoforms produced upon pathogen challenges that are regulated by FPA. The authors will need to provide some of these data to fill this gap.

(2) Since the function of FPA is to regulate NLR immune response genes, we should expect a change in plant defense phenotype in FPA loss-of-function mutants. Could the authors provide more information on this? On the contrary, in line 728 of this manuscript, the authors found that at least for some pathogens, "loss of FPA function does not reduce plant resistance". It is not consistent with the hypothesis that FPA is important to regulate NLR immune response genes.

(3) Furthermore, the authors mentioned in lines 729-731 "Greater variability in pathogen susceptibility was observed in the fpa-8 mutant and was not restored by complementation with pFPA::FPA, possibly indicating background EMS mutations affecting susceptibility." Does it mean that fpa-8 contains other mutations? Will these additional mutations complicate the results of the RNA processing? Could the authors outcross the fpa-8 mutation to a clean background?

(4) In line 318, the authors found 285 and 293 APA events in the fpa-8 mutant and the 35S::FPA:YFP construct respectively, but only 59 loci (line 347) exhibited opposite APA events (about one fifth). The low overlapping frequency suggests that some results could be false positive.

(5) In line 732-736: "In contrast, 35S::FPA:YFP plants exhibited a similar level of sporulation to the pathogen-sensitive Ksk-1 accession (median 3 sporangiophores per plant). This suggests that the premature exonic termination of RPP7 caused by FPA has a functional consequence for Arabidopsis immunity against Hpa-Hiks1." It is contradictory to the statement in line 728 that "loss of FPA function does not reduce plant resistance". Is it possible that overexpression of FPA:YFP had generated an artificial condition that is not related to the natural function of FPA?

(6) The fpa-8 mutant has a delayed flower phenotype (Plant Cell 13:1427). Could the 35S::FPA:YFP fusion protein construct reverse this phenotype and the plant defense response phenotype? It is important to interpret the data when the 35S::FPA:YFP construct was used to represent the overexpression of FPA.

(7) Under the subheading "FPA co-purifies with the mRNA 3' end processing machinery". The results were based on in vivo interaction proteomics-mass spectrometry. MS prompts to false positives and will need proper controls and validations. Have the authors added the control of 35S:YFP instead of just the untransformed Col-0? At least for the putative interacting partners in Figure 1A, could the authors perform validations of some important targets, using techniques such as reverse co-IP, or to show direct protein-protein interaction between FPA to a few of the important targets by in vitro pull-down, BiFC, or FRET, etc.

(8) In Fig. 3, the data show that the last exon of the FPA gene is missing in the FPA transcripts generated from the 35S::FPA:YFP construct. Will the missing of this exon affect the function of the transcript and the encoded protein?

(9) The function of FPA is still ambiguous. There was a quantitative shift toward the selection of distal poly(A) sites in the loss-of-function fpa-8 mutant and a strong shift to proximal poly(A) site selection when FPA is overexpressed (35S::FPA:YFP) in some cases (Fig. 3, Fig. 5, Fig. 8). But the situation could be kind of reversed in other cases (Fig. 6). What is the mechanism behind it?

(10) Under the subheading: "The impact of FPA on NLR gene regulation is independent of its role in controlling IBM1 expression". IBM1 is a common target of FPA and IBM2. Indeed, FPA and IBM2 share several common targets (Plant Physiol. 180:392). It may be more meaningful to compare the impact of FPA and IBM2 on NLR gene instead.

(11) In lines 423-425, the authors described "Consistent with previous reports, the level of mRNA m6A in the hypomorphic vir-1 allele was reduced to approximately 10% of wild-type levels (Parker et al., 2020b; Ruzicka et al., 2017) (Figure 4-figure supplement 3)." This data could not be found.

(12) In line 426: "However, we did not detect any differences in the m6A level between genotypes with altered FPA activity." Which data this statement is referring to?

**Reviewer #3:**

In the article "Widespread premature transcription termination of *Arabidopsis thaliana* NLR genes by the spen protein FPA", the authors describe the function of FPA as a mediator of premature cleavage and polyadenylation of transcripts. They also focused their study on NLR-encoding transcripts, as that was their most novel observation, describing an additional layer of control.

In general, the article is well written and clear. The experimental design is good, they didn't seem to over-interpret the results, the controls were solid, and the nanopore data were quite informative for their work. It is rather descriptive maybe bordering on dry in parts - but the results will be helpful for those working on NLRs, and demonstrate the utility of bulk long-read transcript data. The authors were able to string together a number of descriptive observations or vignettes into an informative paper. Overall, it is solid science, but maybe not monumental.

One minor complaint is that the authors don't focus on NLRs starting on line 436, and then they have extensive results on NLRs; by the time I got to the discussion, I'd forgotten about the early focus on the M6A. While the first part of the article is necessary, I would suggest a more concise results section to give the paper more focus on the NLR control (since that is emphasized in the abstract and the title of the manuscript).

[Editors' note: further revisions were suggested prior to acceptance, as described below.]

Thank you for resubmitting your work entitled "Widespread premature transcription termination of *Arabidopsis thaliana* NLR genes by the spen protein FPA" for further consideration by *eLife*. Your revised article has been evaluated by Detlef Weigel (Senior Editor), Hao Yu (Reviewing Editor) and three reviewers who reviewed the last version of the manuscript.

We feel that this revised manuscript has been significantly improved, but there are some remaining issues that need to be addressed, as indicated in the following comments given by Reviewers 1 and 2.

*Reviewer #1 (Recommendations for the authors):*

The authors made great efforts to reorganize the manuscript to address comments from all three reviewers. Current manuscript supports the main claim on FPA modulating the NLR regulation with a series of graphic illustration as main figures with supporting supplements. These encompass the breadth of regulatory roles of FPA on different NLR genes, in particular. Their quantitative assessment of the FPA effects on clustered or hypervariable NLR genes have been performed in a sound way, taking on the latest research outcomes (2020-2021 publications) on NLR diversity and evolution.

*Reviewer #2 (Recommendations for the authors):*

Overall, it is a piece of interesting research supported with rich data. The authors have addressed much of the concerns in the revised version and through further explanations. Some remaining questions could be addressed via clarification, strengthened comparison, and additional discussions.

1. In relation to my original Question 1. Since the title of this manuscript is "Widespread premature transcription termination of *Arabidopsis thaliana* NLR genes by the spen protein FPA" and some NLR gene expressions are responsive to pathogen attack, the readers may be interested to know the changes in NLR genes under pathogen attack conditions that are regulated by FPA. If the authors have these data, it will be great to share.

2. In relation to my original Question 2 and Question 5. Since overexpression of FPA only partially reduces the level of functional RPP7 transcripts, is it possible that FPA overexpression also acts on other NLR transcripts that leading to loss of resistance?

3. In relation to my original Question 4. Is it possible to make a comparison directly between the 35S::FPA:YFP line versus the fpa-8 mutant to investigate see whether all disappeared pre-mature transcriptional terminations have returned to the level of Col-0 or even more?

4. In relation to my original Question 6. The authors showed that overexpression FPA will decrease the overall FLC transcripts. Is the FPA acting on the pre-mature transcriptional termination of FLC too? Any data to support this?

5. In relation to my original Question 7. Does the anti-FPA chip data match well with the proximal APA in Col-0?

6. In relation to my original Question 9 and Question 10. IBM1 is a common target of FPA and EDM2, indicating the possible coordination of the FPA and EDM2 functions. There have been several studies on EDM2, could the authors compare the target of FPA and EDM2, and also address whether FPA also targets TEs in introns of function genes similar to that of EDM2?

*Reviewer #3 (Recommendations for the authors):*

I am satisfied with the authors' response to the reviewers, including the valuable points raised by the other reviewers. The extensive changes that the authors made to the manuscript have substantially improved the work.

---

## [Author Response]

Essential Revisions:While there was agreement that the topic is timely and the findings relevant, there were some concerns regarding manuscript structure, inconsistency among some results, and interpretation of biological relevance of the data, as listed below, that need to be addressed to support the conclusions.1. The manuscript presents an extensive body of studies in analyzing FPA interacting proteins and its potential RNA targets including NLRs. Although the overall results cover a series of observations, many of them are descriptive and divert the audience's attention from understanding the novelty and significance of the findings. Thus, we suggest that the authors re-organize the manuscript into a more coherent story and focus on the most important data pertaining to NLR control as shown in the title and the abstract.

We have restructured the manuscript to address this concern. Following the specific comments of the reviewers we have:

– Carried out a major-re-write of the Results section.

– Reduced the detailed descriptions in the proteomics and Illumina RNA-seq sections, so as to reach the NLR focus of the study more quickly.

– Created a new main text Figure explaining RNA processing around RPS6.

– Summarized the different FPA-dependent effects on NLRs into 3 new tables to provide a broader view that complements the examples detailed in the text and the detailed data provided as supplementary materials.

– We have removed non-NLR examples of FPA regulation from the manuscript, to focus on the NLR aspect.

– We have changed the text-based descriptions of FPA-NLR control, to rationalize the logic and focus of illustrated examples.

– We have moved the Figure on *IBM1* from the main text to a Supplementary Figure.

– We have edited the section on RPP7-dependent pathogen testing, to clarify apparent misunderstanding.

2. The authors should address or explain some inconsistencies in the results as mentioned by reviewers. For example, the authors found that at least for some pathogens, "loss of FPA function does not reduce plant resistance". This is not consistent with the hypothesis that FPA is important for regulating NLR immune response genes, and the observation that premature exonic termination of RPP7 caused by FPA has a functional consequence for Arabidopsis immunity against Hpa-Hiks1.

There is a straightforward misunderstanding here, possibly because our text in the relevant section was not sufficiently clear.

We tested the impact of different activity levels of Arabidopsis FPA on NLR function by investigating the NLR, RPP7. We chose RPP7 because features of its function and regulation are relatively well characterised. RPP7 provides disease resistance to the oomycete pathogen *Hyaloperonospora arbidopsidis (Hpa)* strain Hiks1. The reference Arabidopsis accession, Col-0, encodes a functional RPP7 gene and hence is resistant to *Hpa-Hiks1* infection. Not all Arabidopsis accessions are resistant to all *Hpa* strains. For example, the Duc-1 and Ksk-1 accessions have been reported as having susceptibility to *Hpa*-Hiks1 infection, likely due to the lack of a functional RPP7 gene (Lai et al., 2019). It was for this reason that we incorporated the Ksk accession as an infectionsensitive positive control accession in our pathogen tests.

The question we were addressing was: Does FPA-dependent premature cleavage and polyadenylation in RPP7 exon 6 compromise RPP7 function? To address this question, we therefore applied *Hpa*-Hiks to our different genetic lines. Neither Col-0 nor the *fpa-8* mutant (which is in the Col-0 genetic background) were sensitive to infection. This is consistent with our hypothesis because the poly(A) site used in exon 6 in Col-0, is used significantly less in *fpa-8*. Hence, there is no compromise in the expression of full-length RPP7 in *fpa-8* mutants. As Col-0 is already resistant to *Hpa*-Hiks1, we would therefore expect *fpa-8* to also be resistant and indeed, this is what we found.

This was also true when we tested an independent allele, *fpa-7,* that is also in the Col-0 background. However, when we tested the line that was over-expressing FPA, which was introduced into an *fpa-8* background (and hence, ultimately Col-0), we found that resistance was lost and *Hpa*-Hiks1 was able to infect these plants.

Therefore, the findings from this experiment are completely consistent “with the hypothesis that FPA is important for regulating NLR immune response genes, and the observation that premature exonic termination of RPP7 caused by FPA has a functional consequence for Arabidopsis immunity against Hpa-Hiks1.” We have clarified the text in this section to make our hypothesis and findings clearer.

3. The significance of this study will be strengthened by analysis of the biological relevancy of the alternative polyadenylation events mediated by FPA pertaining to NLR functions. We suggest that the authors consider either providing new experimental data or clearly interpreting existing results, such as those relevant to regulation of RPP7, to provide better insights into biological significance of the data presented in this manuscript.

To more clearly address the predicted consequences of FPA regulated alternative polyadenylation, we have added tables for the three classes of FPA-regulated alternative polyadenylation events to the main text and made predictions of the functional consequences for intronic and exonic polyadenylation events. Since there are several documented examples of TIR-only or LRR-truncated NLRs regulating resistance and cell death, we have also changed some of the example genes used to focus on those proximal polyadenylation events which can alter protein coding potential.

Please also take into consideration the other specific comments from the reviewers below to revise the manuscript.**Reviewer #1:**[...] If authors wishes to opt for highlighting NLR analysis, the following suggestions would help (9-14).1. Earth mover distance (EMD) has been applied to identify a locus with alternative polyadenylation. What is the basis of using EMD value of 25 as a cutoff? According to Figure 4 B,D, EMD can range from 0-4000. One would also wonder if the distance unit equals bp. In addition, EMD values of some genes (e.g. FPA and representative NLRs) can be specified in the main dataset so that significance of the cut-off values shall be appreciated.

We found that for some very highly expressed loci, we were able to detect statistically significant changes in poly(A) site usage with very small effect sizes which were unlikely to represent functionally important changes. An EMD threshold was therefore required for removing these small effect size loci. The EMD is informally described as the minimum amount of “work” required to turn one distribution into another – it represents the percentage of the distribution moved multiplied by the distance moved. For example, an EMD of 25 could describe a situation where 10% of the transcripts have shifted by 250 nt, or 50% of the transcripts have shifted by 50 nt. A threshold of 25 gives a good trade-off between the percentage of proximal/distal site switching, and the distances between sites (since larger changes in distance are more likely to result in functional changes). We have included EMD values for example NLRs in the main text to give an idea of effect sizes of these genes.

2. Regarding the manual annotation of alternatively polyadenylated NLR genes (L1160-): Genes with alternative polyadenylation were identified and the ending location was supported when there were minimum four DRS reads. It would be relevant to provide the significance of "the four" based on read coverage statistics, for example, with average read number covering an annotated NLR transcript with the specification of an average size.

We have previously demonstrated that both Helicos and Nanopore DRS reads are able to capture the true 3’ ends of single RNA molecules. However, both techniques have some technical limitations which may result in artefacts – for example, the over-splitting of nanopore signal from a single molecule into multiple reads, or the incorrect alignment of low-quality basecalls at the ends of reads. For this reason, and also to standardise our approach to manually identifying FPA-regulated NLRs, we developed a standard operating procedure. We chose to identify poly(A) sites using a minimum of four nanopore read alignments, as a trade-off between sensitively detecting genuine alternative polyadenylation events, and ignoring events caused by poor alignment of low-quality reads or over-splitting. We also looked for evidence of events seen in nanopore data in other sequencing datasets, particularly the Helicos DRS alignments, to corroborate our findings. We have improved the language of the relevant methods section to clarify this.

3. Figure 4E shows that Ilumina-RNAseq dataset detects the number of loci with a different order of magnitude compared with the other two methods. Reference-agonistic pipeline shall be appreciated, however, the method engaged might have elevated the counting of paralogous reads mapped to different locations than they should be. Along with paralogous read collapsing, this is always a problem with tandemly repeated genes, such as NLRs by and large. For example, NLR paralogs in a complex cluster with conserved TIR/NBS but diversified LRRs would have higher coverage in the first two domains but drop in the diversified parts. The authors need to specify their bioinformatic consideration to avoid such problems.

Although the tone of the Illumina read section was careful and the main 3'-end processing conclusion was made by nanopore DRS, the authors are also advised to clearly state the limitation of using Illumina-RNAseq to address alternative polyadenylating sites at the beginning of the section, for example what to be maximally taken out from Figure 4 E and 4F. This will give relative weights to each dataset generated by different methods. One advantage of using Illumina data would be that the expression level changes can be associated with changes in processing, it seems.

The reviewer is correct that multimapping reads are an issue at NLR genes and may lead to uneven coverage of uniquely and multi-mapped reads when some regions of a gene are divergent, and others are not. Although it is the relative change in coverage of exons or expressed regions which is important in DEXSeq analysis (rather than absolute coverage), it is possible that changes in processing that cause relative expression changes at one NLR locus may have impacts on the relative expression of multimapping regions at other paralogous NLR loci. We addressed this issue when quantifying the expression of expressed regions by running featureCounts using the –primary option that only counts primary alignments, but we failed to mention this in the methods. We have updated the methods to clarify this.

4. At the RPP7 locus, At1g58848 is identical in sequences with At1g59218 as is At1g58807 with At1g59214 (two twins in the RPP7 cluster by tandem duplication). It would be good to check whether the TE At1g58889 readthrough indeed occurs in the sister duplicate with a potential TE in the downstream of At1g59218. If not, it can be used as an example of duplication and neofunctionalization through an alternative polyadenylation site choices.

The tandem duplication of AT1G58848 and AT1G58807 in Col-0 makes the *RPP7* locus complex to analyse even with long read sequencing data. We find that even with nanopore DRS data, nearly all reads mapping to AT1G58807 multimap at AT1G59124. There is clear evidence of exonic proximal polyadenylation in these transcripts, but the locus of origin is not determinable. In the case of AT1G58848 and AT1G59218, we find a mixture of uniquely mapping and multimapping reads at both genes, and both genes have uniquely mapped reads indicating exonic proximal polyadenylation in *35S::FPA*, and chimeric RNA formation in *fpa-8*. This suggests that RNA processing of these loci is very similar, and so we opted only to show AT1G58848 as an example. Due to the much shorter length of Helicos DRS reads, we applied much more stringent filtering to remove incorrectly mapping or multimapping reads, meaning that there were not enough uniquely mapped reads at the AT1G58848 and AT1G58807 loci to perform Helicos EMD tests. We have updated the text to explain this more clearly.

5. HMM search shall be revisited to confirm if they are to detect the TIR domain. Given that a large proportion of NLRs in *A. thaliana* carry TIR at their N-terminal ends and the specified examples included TIR-NLR, it is surprising to see no TIR domain in Figure 5.

The absence of the Interpro annotation from Figure 5C (now Figure 4A in the revised manuscript) is a mistake on our part rather than due to its absence from the Interpro annotation. We have now corrected the figure and all other gene tracks to make sure that all Interpro annotations are shown.

6. L659-668: how does the new data relate to the previously TAIR annotated At1g58602.1 vs At1g58602.2 (Figure 6, Inset 1)? It would be good to see these clearly stated in the main text as compared to newly identified ones. From the nanopore profiling, At1g58602.2 appears to be the dominant form.

AT1G58602.2 from the Araport11 annotation contains the most distal annotated isoform of *RPP7*, whilst AT1G58602.1 contains a slightly more proximal 3’UTR. The reviewer is correct that AT1G58602.2 is the more dominant isoform in our Col-0 data. We have added a sentence that acknowledges this to the section on *RPP7* 3’UTR isoforms.

7. One thing to note is that in the overexpressor of which Hiks1 R is suppressed, there was hardly any At1g58602.1 produced in addition to the large reduction of At1g58602.2. Thus, relative functional importance of the two transcripts shall be discussed in line with the Hpa resistance data. Accordingly, L740-741 phrasing shall be revised to include the possibility of absolute or relative "depletion" of functional transcript(s) contributing to the compromise in Hpa resistance.

While we agree that, in principle, the change in relative expression of the two annotated distal isoforms of *RPP7* could have functional consequences, given that both of these isoforms can encode a protein, the functional impact of this relative change is much less likely to be the cause of the loss of Hpa resistance in FPA overexpressing plants, compared to the larger change in exonic proximal polyadenylation, which produces transcripts which are unlikely to express protein. Given that we have not demonstrated conclusively that it is the increase in exonic polyadenylation of *RPP7* that causes reduced immunity in *35S::FPA:YFP*, we have made the language of our conclusions in the section “FPA modulates RPP7-dependent, race-specific pathogen susceptibility” more careful.

8. It would be necessary to state in the main text the implication of phosphorylation on the two Ser residues on Pol II at L245. A clear description distinguishing the effect of the two phosphorylation and the specificity of the antibodies is desirable, as the data was interpreted as if the two sites made differences, such that Ser2 was heavily emphasized (e.g. subtitle). Albeit low level, Ser5 data also shows an overlap with FPA ChIP-seq coverage at the 3' end. If there is a statistical significance to be taken account to interpret the coverage, please state it. Given that elongation occurs progressively, I wonder how much should be taken out from the distinction.

It is well established in the literature that Pol II phosphorylated at Ser^5^ of the C-terminal domain is a hallmark of initiating and elongating Pol II, whilst Ser^2^ is a hallmark of terminating Pol II (Phatnani and Greenleaf, 2006). This was first established in yeast, where it was shown that Ser5 phosphorylation is necessary for the recruitment of the mRNA capping machinery (Cho et al., 1997; Ho and Shuman, 1999). The yeast homolog of 5’-to-3’ exonuclease which is required for termination (West et al., 2004), was also shown to interact specifically with Pol II phosphorylated at Ser^2^ via an accessory protein (Kim et al., 2004). Therefore, comparing FPA occupancy to relative levels of Ser^2^ and Ser^5^ phosphorylated Pol II is an important validation of the location of FPA binding. We have added a sentence to the relevant Results section describing why CTD phosphorylation varies through the gene body. Arabidopsis ChIP-seq experiments from the literature which profile all Pol II (not just phosphorylated versions) indicate that in Arabidopsis, the highest occupancy is over the terminator (Yu et al., 2019). This may explain why there is also a peak of Ser^5^ at the terminator (i.e. if there are low levels of Ser^5^ in a region of higher occupancy, or if there is cross-reactivity of the antibody with Ser^52^ or unphosphorylated Pol II).

9. Figures presentation for RPP4 and RPP7 are great in detailing the FPA-dependent NLR transcript complexity. To make the functional link more evident, the authors may consider bringing up parts of the Figure 5-supplement to a main Figure to detail the revised annotation of NLRs. Given recent advances in NLR structure and function studies, extra domain fusion, fission and truncated versions of NLRs require a great deal of attention. For example, potential functional link to the NMD-mediated autoimmunity and revised annotation of At5g46470 (RPS6) needs a clear visual guidance preferably with a main figure (Figure 5-Figure Supplement 3).

We thank the reviewer for this comment, and we agree that these figures deserve to be made more visible. This is one of the reasons that we have chosen to submit our manuscript to eLife, since supplementary figures are displayed alongside linked main text figures in an image slider which allows easy access to each gene track. We believe that this will also make it much easier to examine individual gene tracks, without having to compress them to fit them into a single figure panel. However, we do agree that RPS6 is particularly interesting and deserves to be a main figure. We have therefore split the NLR figure into two new figures and incorporated *RPS6* gene tracks into the first of these.

10. The section "FPA controls the processing of NLR transcripts" includes dense information and can be broken down to several categories. To this end, Supplement File 3 (NLR list) shall be revised to deliver the categorical classes and further details and converted to a main table.For NLR audience, for example, it would be important to associate the information to raw reads to assess where the premature termination would occur. At least, the ways to retrieve dataset or to curate the termination sites shall be guided.On the contrary, there is no need to include other genes in Figure 4-Figure Supplement 4-8 under this section. They are not NLRs.

We have created main-text tables for each of the three classes of FPA-regulated NLR genes, as suggested by the reviewer. We have also removed the examples of non-NLR genes regulated by FPA from the paper, to streamline the story. All the datasets analysed in the study are already available on ENA with database identifiers provided in the Data Availability section to guide readers.

11. Figure 7 and IBM1 section can be spared to the supplement.

We have followed the reviewer’s suggestion and this figure now appears as Figure 2 supplement 4. We have moved the results section on IBM1 up to join it with the global analysis of FPA function in RNA processing.

12. The list of "truncated NLR transcripts" in particular, either by premature termination within protein-coding or with intronic polyadenylation, should be made as a main table. The table can be preferably carrying details in which degree the truncation is predicted to be made. With current sup excel files, it is difficult to assess the breadth of the FPA effect on the repertoire of NLRs and their function. This way, functional implication of differential NLRs transcriptome can be better emphasized.

We have followed the reviewer’s suggestion here and prepared this information into main-text tables 1-3, including predictions of the functional consequences for intronic/exonic poly(A) site choice.

13. FPA-mediated NLR transcript controls, as to promote transcript diversity, is expected to exert its maximum effect if FPA level or activity is subject to the environmental stresses, such as biotic or abiotic stresses. The discussion on effectors targeting RNA-binding proteins (L909-918) is a great attempt in broadening the impact of this research. In addition, if anything is known to modulate FPA activity, such as biotic or abiotic stresses or environmental conditions, please include in the discussion.

We are not aware of any literature reporting the modulation of FPA activity by biotic or abiotic stresses. This is certainly an interesting question which we would like to examine. However, the analysis of FPA activity is complicated by a number of factors. RNA-level expression is often used as a proxy for overall activity. The RNA-level expression of FPA is not necessarily indicative of FPA activity, however, since the proximally polyadenylated isoform of FPA does not produce functional FPA protein. To get a clear picture of FPA activity during infection will therefore require high-depth Illumina RNA-Seq, nanopore direct RNA sequencing or proteomics analysis.

14. NLR transcript diversity as source of cryptic variation contributing to NLR "evolution" is an interesting concept, however, evolutionary changes require processes of genic changes affecting transcript layers or stabilizing transcriptome diversity. In the authors' proposition in looking into accessions, potential evolutionary processes can be further clarified.

We agree with the reviewer that a species-wide transcriptome analysis would provide an invaluable insight into how transcription can affect evolutionary changes. For example, we find that NLRs with high levels of allelic diversity are more likely to be regulated by proximal polyadenylation in Col-0, and so a species-wide approach will reveal whether this regulation is conserved or tailored to environmental conditions. An integrative analysis of genomic and transcriptomic data will also help to identify whether chimeric RNAs present in some accessions are found as retrotransposed genes in others. We have added these specific example experiments to the relevant discussion section.

**Reviewer #2:**[...] Overall, it is a potentially important research. The data is rich and could be useful. However, the biological stories described are not thoroughly supported by the data presented, especially when the authors tried to touch on several aspects without some important validations and strong connections among different parts. Some special comments are provided below:1. The title of this manuscript is "The expression of Arabidopsis NLR immune response genes is modulated by premature transcription termination and this has implications for understanding NLR evolutionary dynamics". Therefore, the readers will expect some functional connections between the FPA and the novel NLR isoforms due to premature transcription termination. However, the transcript levels of plant NLR genes are under strict regulation (e.g. Mol. Plant Pathol. 19:1267). Since the functions of NLR genes are related to effector-triggered immunity, it is more important to study the function of FPA on premature transcription termination when the plants are challenged with pathogens. In this manuscript, most transcript analyses are based on samples under normal growth conditions. It is therefore a weak link between the genomic studies and the functional aspects. For instance, it is more important to identify unique NLR isoforms produced upon pathogen challenges that are regulated by FPA. The authors will need to provide some of these data to fill this gap.

To clarify, the title of this manuscript is not as stated here by the reviewer but is “Widespread premature transcription termination of *Arabidopsis thaliana* NLR genes by the spen protein FPA”. We do indeed describe a functional pathogen test to examine the functional impact of FPA. We show that overexpression of FPA reduces the functional expression of RPP7 transcripts, and that this impacts upon the ability of plants to resist Hpa-hiks1. We agree with the referee that it will be very interesting to investigate, not just FPA, but changes in 3’ processing during infection by different pathogens. However, key questions on NLRs extend to how they function, how they evolve, how they trigger hyperimmunity and how they are controlled to limit impact on fitness, all of which may be impacted by the control of RNA 3’ processing.

2. Since the function of FPA is to regulate NLR immune response genes, we should expect a change in plant defense phenotype in FPA loss-of-function mutants. Could the authors provide more information on this? On the contrary, in line 728 of this manuscript, the authors found that at least for some pathogens, "loss of FPA function does not reduce plant resistance". It is not consistent with the hypothesis that FPA is important to regulate NLR immune response genes.

There is a misunderstanding here, which we clarify in response to Essential Revisions 2.

3. Furthermore, the authors mentioned in lines 729-731 "Greater variability in pathogen susceptibility was observed in the fpa-8 mutant and was not restored by complementation with pFPA::FPA, possibly indicating background EMS mutations affecting susceptibility." Does it mean that fpa-8 contains other mutations? Will these additional mutations complicate the results of the RNA processing? Could the authors outcross the fpa-8 mutation to a clean background?

Given that the *fpa-8* mutant was generated using EMS treatment, it is probable that it does contain other mutations besides the one that removes FPA function (this is likely to be the case with most mutants – whether they are generated with EMS or T-DNA insertions). These mutations are likely to be the source of the slightly greater variability in susceptibility to *Hpa-hiks1* in *fpa-8* compared to the *fpa-7* T-DNA mutant. These potential off-target mutations are unlikely to be the cause of the RNA 3’ processing changes seen in the *fpa-8* mutant, however, for three reasons: (i) we have previously published Helicos DRS data from *fpa-7* mutants which shows that they have the same RNA 3’ processing defects as *fpa-8* mutants, for example at *PIF5* and *IBM1* (Duc et al., 2013) indicating that changes in 3’ processing in *fpa-8* and *fpa-7* are caused by the common loss of FPA function; (ii) our Illumina RNA-Seq data for the FPA complementing line shows that an FPA transgene restores 3’ processing effects seen in the *fpa-8* mutant, for example at *PIF5* (Author response image 1), but does not restore the variability in susceptibility of *fpa-8* to Hpa-hiks1 (Figure 6C) (iii) many of the genes with altered poly (A) site choice in *fpa-8*, including *RPP7*, show reciprocal changes in processing in the FPA overexpressing line. Taken together, these findings strongly indicate that the loss of FPA is what causes altered poly (A) site choice in an *fpa-8* mutant.

**Author response image 1. respfig1:** A *pFPA::FPA* transgene complements chimeric RNA formation found in the *fpa-8* mutant at *PIF5*. Illumina RNA-Seq data showing the expression of *PIF5* (AT3G59060) - *PAO3* (AT3G59050) chimeric RNAs in *fpa-8* is lost in *pFPA::FPAfpa-8* complemented lines.

4. In line 318, the authors found 285 and 293 APA events in the fpa-8 mutant and the 35S::FPA:YFP construct respectively, but only 59 loci (line 347) exhibited opposite APA events (about one fifth). The low overlapping frequency suggests that some results could be false positive.

The level of reciprocal alternative polyadenylation cannot be used to determine false positive rate. For a gene to show reciprocal effects, when comparing the results of *fpa-8* vs Col-0, and *35S::FPA:YFP* vs Col-0, requires at least two poly(A) sites to be used at high levels in Col-0. For example, at *RPP7*, high levels of proximal exonic polyadenylation are detectable in Col-0, meaning that a shift to distal site usage is detectable in *fpa-8*, as well as the shift to proximal site selection in *35S::FPA:YFP*. However, there are many loci where this is not the case. For example, the abundant chimeric RNAs found at the *PIF5* locus in *fpa-8* are undetectable in Col-0, meaning that overexpression of FPA has no effect on PIF5 when compared to Col-0. Consequently, *PIF5* is not amongst those genes with reciprocal regulation, despite the effect of FPA on *PIF5* RNA processing being very clear in multiple datasets.

5. In line 732-736: "In contrast, 35S::FPA:YFP plants exhibited a similar level of sporulation to the pathogen-sensitive Ksk-1 accession (median 3 sporangiophores per plant). This suggests that the premature exonic termination of RPP7 caused by FPA has a functional consequence for Arabidopsis immunity against Hpa-Hiks1." It is contradictory to the statement in line 728 that "loss of FPA function does not reduce plant resistance". Is it possible that overexpression of FPA:YFP had generated an artificial condition that is not related to the natural function of FPA?

There is a misunderstanding here that may be due to the wording that we used in this section and we explain this above. Col-0 is resistant to *Hpa-Hiks1* because it has a functional *RPP7* gene. In *fpa-8* mutants, the expression of full-length RPP7 transcripts is not compromised relative to Col-0 and hence it is as resistant to *Hpa-Hiks1* as Col-0. In contrast, *35S::FPA:YFP* promotes the use of a poly(A) site within exon 6, reducing the amount of full-length RPP7 detected. This poly(A) site is used in the Col-0 wildtype line but is not detectably selected in the loss-of-function *fpa-8* mutant line. Together, these findings reveal that this poly(A) site is chosen in the Col-0 reference strain and that this requires FPA. Therefore, the selection of this site is the natural function of FPA and not simply generated by an artificial condition. We have re-worded the text in this section to clarify this misunderstanding.

6. The fpa-8 mutant has a delayed flower phenotype (Plant Cell 13:1427). Could the 35S::FPA:YFP fusion protein construct reverse this phenotype and the plant defense response phenotype? It is important to interpret the data when the 35S::FPA:YFP construct was used to represent the overexpression of FPA.

As we report in the Materials & Methods section, a line expressing *35S::FPA:YFP* was obtained from Caroline Dean. Published evidence that this line complements the late flowering phenotype of *fpa-8* is provided in the corresponding publication (Baurle et al., 2007) as Figure S5. In our growth conditions, these lines flower early like wild-type compared to the very late flowering of *fpa-8*. The late flowering phenotype of *fpa-8* mutants is explained by elevated levels of the floral repressor FLC. The Illumina RNA-Seq, Helicos DRS and nanopore DRS data that we release here all show reduced levels of *FLC* in the *35S::FPA:YFP* line compared to *fpa-8* consistent with complementation (Author response image 2).

**Author response image 2. respfig2:** A *35S::FPA:YFP* transgene complements elevated expression of FLC found in the *fpa-8* mutant. Illumina RNA-Seq data showing the overexpression of FLC (AT5G10140) in *fpa-8* is restored to around wild type levels in *pFPA::FPA* and *35S::FPA:YFP* complemented lines.

7. Under the subheading "FPA co-purifies with the mRNA 3' end processing machinery". The results were based on in vivo interaction proteomics-mass spectrometry. MS prompts to false positives and will need proper controls and validations. Have the authors added the control of 35S:YFP instead of just the untransformed Col-0? At least for the putative interacting partners in Figure 1A, could the authors perform validations of some important targets, using techniques such as reverse co-IP, or to show direct protein-protein interaction between FPA to a few of the important targets by in vitro pull-down, BiFC, or FRET, etc.

FP fusions are widely used in IP experiments, but we are not aware of any study that reports 3’ processing factors to be recurrent contaminants in such experiments. We had anticipated submitting an additional proteomics study at around the same time as this study but aspects of this additional work were disrupted by control measures associated with Covid-19. What we do show here, is that an orthogonal approach (ChIP) with different antibodies (anti-FPA) also localises FPA to the 3’ end of Arabidopsis genes together with Pol II phosphorylated on Ser2 of the CTD. These orthogonal datasets are therefore consistent with our interpretation that FPA co-purifies with Pol II and multiple factors involved in the processing of RNA 3’ ends and are also supported by our transcriptomic analyses of *fpa* mutants and overexpressors which have altered 3’ processing.

8. In Fig. 3, the data show that the last exon of the FPA gene is missing in the FPA transcripts generated from the 35S::FPA:YFP construct. Will the missing of this exon affect the function of the transcript and the encoded protein?

As we state in the Materials & Methods section, this line was obtained from Caroline Dean and the details of its construction were previously described (Baurle et al., 2007). The transgene construct has a different promoter (CaMV 35S) and associated 5’UTR sequence and the sequence downstream of the stop codon is replaced by a transgene-derived 3’UTR. Consequently, these regions of the transgene-derived *FPA* do not align to the Col-0 reference. We have added new text to the Figure legend to clarify this point. Given that the *35S::FPA:YFP* transgene complements the flowering time phenotype of *fpa-8* mutants, and causes widespread changes in 3’ processing, there is no evidence that the lack of the canonical 3’UTR has a deleterious impact on the function of the FPA protein.

9. The function of FPA is still ambiguous. There was a quantitative shift toward the selection of distal poly(A) sites in the loss-of-function fpa-8 mutant and a strong shift to proximal poly(A) site selection when FPA is overexpressed (35S::FPA:YFP) in some cases (Fig. 3, Fig. 5, Fig. 8). But the situation could be kind of reversed in other cases (Fig. 6). What is the mechanism behind it?

Using different sequencing technologies, we clearly show that the predominant effect of FPA is to promote proximal poly(A) site selection and indeed that these cases are associated with the largest effect sizes. The mechanism involved is not studied here. One possibility is that genes which display an increase in distal polyadenylation when FPA is overexpressed are indirect targets of FPA. This would be unsurprising given that FPA regulates the alternative polyadenylation of a number of other factors involved in 3’ processing. Another possibility is that FPA can associate with different complexes of 3’ processing factors at different locations, resulting in opposing effects on 3’ processing. A future goal for us, in dissecting the mechanism by which FPA mediates NLR transcription termination will be to relate poly(A) site choice to direct RNA binding site interactions mapped by iCLIP, for example.

10. Under the subheading: "The impact of FPA on NLR gene regulation is independent of its role in controlling IBM1 expression". IBM1 is a common target of FPA and IBM2. Indeed, FPA and IBM2 share several common targets (Plant Physiol. 180:392). It may be more meaningful to compare the impact of FPA and IBM2 on NLR gene instead.

IBM2/ASI1 is an RNA and chromatin binding protein that regulates the expression of IBM1 by promoting elongation through intronic heterochromatic marks, as part of a complex with EDM2 and AIPP1. As a result, *edm2*, *ibm2*, and *aipp1* mutants fail to produce full length *IBM1* transcripts, resulting in phenotypes similar to the *ibm1* mutant. Mutations in FPA were recently identified as suppressors of the phenotypes of *ibm2* mutants. This is likely because FPA promotes the proximal polyadenylation of *IBM1* transcripts.

Since FPA regulates the proximal polyadenylation of *IBM1*, we asked if it was possible that some of the targets of FPA overexpression identified by nanopore and Helicos DRS were caused by indirect effects on chromatin state resulting from a decrease in full length IBM1 expression. However, there is no indication that FPA acts to promote alternative polyadenylation of *IBM2*. We therefore consider it unlikely that proximal polyadenylation of NLRs in the *35S::FPA:YFP* line is caused by indirect effects on *IBM2*.

11. In lines 423-425, the authors described "Consistent with previous reports, the level of mRNA m6A in the hypomorphic vir-1 allele was reduced to approximately 10% of wild-type levels (Parker et al., 2020b; Ruzicka et al., 2017) (Figure 4 - supplement 3)." This data could not be found.

We have re-checked the submitted article. These data are indeed there: page 46, line 1510 and correctly labelled as Figure 4 supplement 3. In the revised manuscript these data are included as Figure 2-figure supplement 3, and the raw data is also available as Figure 2 source data 11.

12. In line 426: "However, we did not detect any differences in the m6A level between genotypes with altered FPA activity." Which data is this statement referring to?

This statement refers to the data in Figure 2-figure supplement 3 of the revised manuscript.

**Reviewer #3:**[...] One minor complaint is that the authors don't focus on NLRs starting on line 436, and then they have extensive results on NLRs; by the time I got to the discussion, I'd forgotten about the early focus on the M6A. While the first part of the article is necessary, I would suggest a more concise results section to give the paper more focus on the NLR control (since that is emphasized in the abstract and the title of the manuscript).

We thank the reviewer for their comments. We agree that the paper is dichotomous due to the initial focus on the function of FPA and subsequent identification of the effect on NLRs. We have reduced the length of the initial results sections, particularly the proteomics results, so as to come to our findings on NLR genes more quickly.

[Editors' note: further revisions were suggested prior to acceptance, as described below.]Reviewer #2 (Recommendations for the authors):Overall, it is a piece of interesting research supported with rich data. The authors have addressed much of the concerns in the revised version and through further explanations. Some remaining questions could be addressed via clarification, strengthened comparison, and additional discussions.

We thank the reviewer for these remarks. We have addressed their questions below.

1. In relation to my original Question 1. Since the title of this manuscript is "Widespread premature transcription termination of *Arabidopsis thaliana* NLR genes by the spen protein FPA" and some NLR gene expressions are responsive to pathogen attack, the readers may be interested to know the changes in NLR genes under pathogen attack conditions that are regulated by FPA. If the authors have these data, it will be great to share.

The question of whether FPA (or other RNA binding proteins) alter the 3’ processing of NLR transcripts during infection is something that we would like to explore. Whilst some microarray and RNA sequencing datasets collected during infection conditions are available, these are generally designed for analysing expression changes at the gene level. As a result, they are underpowered for the analysis of RNA processing, which generally requires higher sequencing depth and longer reads – for example 50-100 million 2 x 150bp reads per replicate, with 6 or more biological replicates, as was used in our Illumina experiment. As far as we are aware, no experiments using nanopore direct RNA sequencing to identify RNA processing changes during infection have yet been generated or published. It is clear that the absence of detailed analyses of *NLR* transcript processing and fate during pathogen infection represents a gap in understanding for the immunity field as a whole, and so generating these data is a goal for our future enquiries.

2. In relation to my original Question 2 and Question 5. Since overexpression of FPA only partially reduces the level of functional RPP7 transcripts, is it possible that FPA overexpression also acts on other NLR transcripts that leading to loss of resistance?

We cannot rule out the possibility that FPA-dependent proximal polyadenylation at other NLR loci besides RPP7 may contribute quantitatively to the loss of immunity to *Hpa-*hiks1 seen in the *35S::FPA:YFP* line. We have therefore reworded our conclusions in the relevant Results section. We now state: “We conclude that FPA control of poly(A) site selection can modulate NLR function, with a functional consequence for immunity.”

3. In relation to my original Question 4. Is it possible to make a comparison directly between the 35S::FPA:YFP line versus the fpa-8 mutant to investigate see whether all disappeared pre-mature transcriptional terminations have returned to the level of Col-0 or even more?

We compared *fpa-8* and *35S::FPA:YFP* nanopore DRS data directly using the Earth mover distance (EMD) method, as suggested by the reviewer. We found that 80.0% of the loci with significantly increased distal poly(A) site choice in *fpa-8* when compared with Col-0, were also significant when compared to *35S::FPA:YFP* (hypergeometric *p* = 2.2 x 10^-172^). This indicates that the *35S::FPA:YFP* transgene is able to reverse the readthrough at these loci displayed in *fpa-8*. Furthermore, 77.2% of loci with significantly increased proximal poly(A) site choice in *35S::FPA:YFP* when compared to Col0, were also significant when compared to *fpa-8* (hypergeometric *p* = 1.9 x 10^-119^). Of the loci with altered poly(A) site choice when comparing *35S::FPA:YFP* to either Col-0 or *fpa-8,* 85.9% had a larger EMD when compared to *fpa-8,* suggesting that there are reciprocal changes beyond Col-0 levels in *35S::FPA:YFP* and *fpa-8* at these loci.

4. In relation to my original Question 6. The authors showed that overexpression FPA will decrease the overall FLC transcripts. Is the FPA acting on the pre-mature transcriptional termination of FLC too? Any data to support this?

There is no evidence that sense *FLC* transcripts are targeted by FPA-dependent proximal polyadenylation (Duc et al., 2013; Hornyik et al., 2010). Instead, there is a significant body of literature on the role of FPA and other 3’ processing factors influencing long non-coding antisense RNAs at the *FLC* locus (Hornyik et al., 2010). The ratio of proximal to distal antisense RNAs correlates negatively with sense *FLC* expression (Duc et al., 2013; Hornyik et al., 2010).

5. In relation to my original Question 7. Does the anti-FPA chip data match well with the proximal APA in Col-0?

To test whether there was an overlap between the sites of FPA-associated chromatin and loci with FPA-sensitive poly(A) site choice, we called peaks from the FPA ChIP-seq data using MACS2 (Zhang et al., 2008). This resulted in the identification of 1120 unstranded peaks. We then assigned peaks to loci by identifying the closest (or overlapping) transcribed locus in an upstream orientation to each peak, using bedtools (Quinlan and Hall, 2010). Where there were multiple tied loci that could be assigned to a single peak (e.g. at convergent terminators or overlapping loci), peaks were assigned to all tied loci. We then compared the loci with identified FPA peaks to those loci with FPA-dependent alternative polyadenylation, identified using Nanopore DRS sequencing. We found that of the 222 loci with increased distal polyadenylation in *fpa-8,* 38 were also associated with an FPA ChIP-seq peak (hypergeometric *p* = 3.5 x 10^-4^). Of the 166 loci with increased proximal polyadenylation in *35S::FPA:YFP,* 27 were also associated with an FPA ChIP-seq peak (hypergeometric *p* = 3.3´10^-3^). The lack of FPA ChIP-seq peaks on many genes with FPA-dependent RNA processing may be explained by low Pol II occupancy. In agreement with this, loci with FPA-dependent alternative polyadenylation, which did not have an associated FPA peak, were more weakly expressed than those that did have an FPA peak (Mann-Whitney U *p=*7.2 x 10^-4^). Notably, 95.6% of genes which were associated with FPA ChIP-seq peaks did not have FPA-sensitive alternative polyadenylation under our experimental conditions, and global FPA ChIP-seq signal at 3’ ends was well correlated with Pol II Ser^2^ signal (Spearman’s ρ = 0.67, *p* < 2´10^-308^, 95% CIs [0.66, 0.68]). This suggests that FPA is able to associate with terminating Pol II at most loci but is necessary for poly(A) site choice at a relatively smaller number of loci in our experimental conditions. We have added our findings on the correlation of FPA and Pol II ChIP-seq signal to the relevant Results section and updated the Discussion section “*Uncovering protein assemblies that mediate 3ʹ end processing in living plant cells*”. We now state:

“Such interactions [between FPA and Pol II Ser^2^] could account for the global correlation between FPA and Pol II Ser^2^ occupancy and explain how FPA is able to associate with terminating Pol II at the 3’ ends of most expressed genes.”

6. In relation to my original Question 9 and Question 10. IBM1 is a common target of FPA and EDM2, indicating the possible coordination of the FPA and EDM2 functions. There have been several studies on EDM2, could the authors compare the target of FPA and EDM2, and also address whether FPA also targets TEs in introns of function genes similar to that of EDM2?

Previous studies have indicated that FPA and the EDM2/IBM2/AIPP1 complex act antagonistically to regulate the expression of IBM1 (Deremetz et al., 2019). As a result, mutations disrupting FPA were identified in a genetic screen to isolate suppressors of the *ibm2* mutation, the phenotype of which is caused by dysregulation of *IBM1*. This finding is consistent with our previous discovery that FPA controls the proximal polyadenylation of *IBM1* (Duc et al., 2013). Other genes regulated by EDM2/IBM2/AIPP1 include *RPP7*, *RPP4*, *AT3G05410*, and *AT1G11270* (Lai et al., 2020; Wang et al., 2013). We have shown in this manuscript that FPA-dependent alternative polyadenylation of *RPP7* occurs via an independent exon 6 poly(A) site to the one regulated by EDM2 (which is in intron 1). At *AT3G05410,* loss of EDM2 function causes proximal polyadenylation near the 5’ splice site of intron 3, resulting in complete loss of expression of the distal exon. We find that loss of FPA causes a slight increase in distal polyadenylation of *AT3G05410,* including at a poly(A) site in a cryptic final exon in intron 3 (Author response image 1). This may explain the findings of Deremetz et al., who showed that loss of FPA function partially rescued the proximal polyadenylation of *AT3G05410* in *ibm2* mutants (Deremetz et al., 2019). At *AT1G11270*, loss of EDM2 function causes proximal polyadenylation near the 5’ splice site of intron 3, also resulting in complete loss of expression of the distal exon (Duan et al., 2017). In comparison, loss of FPA function does not have a strong effect on *AT1G11270,* though it may cause a slight increase in intronic proximal polyadenylation at a cryptic final exon in intron 3 (Author response image 3). This suggests that any control of *AT1G11270* by FPA occurs via an independent mechanism to the regulation by EDM2/IBM2/AIPP1. Finally, at *RPP4,* loss of EDM2 function suppresses readthrough into the downstream COPIA retrotransposon (Lai et al., 2020). Similar suppression of readthrough is seen in the *fpa-8* mutant (Figure 5) indicating that at *RPP4*, EDM2 and FPA do not act antagonistically. FPA also controls poly(A) site choice a large number of genes which have no evidence of intronic transposons or intragenic heterochromatin. In addition, FPA predominantly co-purifies with RNA 3’ processing factors rather than histone readers in our proteomics dataset. None of the known interactors of EDM2 and IBM2, which include AIPP1, AIPP2, AIPP3, and CPL2 (Duan et al., 2017), were found associated with FPA. We conclude that regulation of 3’ processing by FPA occurs independently to EDM2 regulated 3’ processing, at a different set of poly(A) sites, and using different mechanisms.

**Author response image 3. respfig3:** Alternative polyadenylation of genes with intronic heterochromatin in *fpa-8* and *35S::FPA:YFP* lines. (A-B) Gene track showing poly(A) site choice of (A) AT3G05410 and (B) AT1G11270 in *fpa-8* and *35S::FPA:YFP* lines.